# Gut microbiota density influences host physiology and is shaped by host and microbial factors

Eduardo J Contijoch[1,2], Graham J Britton[1,2], Chao Yang[1,2], Ilaria Mogno[1], Zhihua Li[1], Ruby Ng[1], Sean R Llewellyn[1], Sheela Hira[3], Crystal Johnson[4], Keren M Rabinowitz[5,6], Revital Barkan[5], Iris Dotan[5,7], Robert P Hirten[8], Shih-Chen Fu[2], Yuying Luo[8], Nancy Yang[8], Tramy Luong[2], Philippe R Labrias[2], Sergio Lira[1], Inga Peter[2], Ari Grinspan[8], Jose C Clemente[1], Roman Kosoy[2], Seunghee Kim-Schulze[9], Xiaochen Qin[1], Anabella Castillo[8], Amanda Hurley[2], Ashish Atreja[8], Jason Rogers[8], Farah Fasihuddin[8], Merjona Saliaj[8], Amy Nolan[8], Pamela Reyes-Mercedes[8], Carina Rodriguez[8], Sarah Aly[8], Kenneth Santa-Cruz[8], Lauren Peters[10,11], Mayte Suárez-Fariñas[2], Ruiqi Huang[2], Ke Hao[2], Jun Zhu[2], Bin Zhang[2], Bojan Losic[2], Haritz Irizar[2], Won-Min Song[2], Antonio Di Narzo[2], Wenhui Wang[2], Benjamin L Cohen[8], Christopher DiMaio[8], David Greenwald[8], Steven Itzkowitz[8], Aimee Lucas[8], James Marion[8], Elana Maser[8], Ryan Ungaro[8], Steven Naymagon[8], Joshua Novak[8], Brijen Shah[8], Thomas Ullman[8], Peter Rubin[8], James George[8], Peter Legnani[8], Shannon E Telesco[12], Joshua R Friedman[12], Carrie Brodmerkel[12], Scott Plevy[12], Judy H Cho[8], Jean-Frederic Colombel[8], Eric E Schadt[11], Carmen Argmann[2], Marla Dubinsky[13], Andrew Kasarskis[2], Bruce Sands[8], Jeremiah J Faith[1]*

[1]Precision Immunology Institute, Icahn School of Medicine at Mount Sinai, New York, United States; [2]Icahn Institute for Genomics and Multiscale Biology, Icahn School of Medicine at Mount Sinai, New York, United States; [3]Zoo Knoxville, Knoxville, United States; [4]Center for Comparative Medicine and Surgery, Icahn School of Medicine at Mount Sinai, New York, United States; [5]Division of Gastroenterology, Rabin Medical Center, Petah Tikva, Israel; [6]Felsenstein Medical Research Center, Sackler Faculty of Medicine, Tel Aviv University, Tel Aviv, Israel; [7]Sackler Faculty of Medicine, Tel Aviv University, Tel Aviv, Israel; [8]The Dr. Henry D Janowitz Division of Gastroenterology, Icahn School of Medicine at Mount Sinai, New York, United States; [9]Division of Hematology and Medical Oncology, The Tisch Cancer Institute, Icahn School of Medicine at Mount Sinai, New York, United States; [10]Department of Genetics and Genomic Sciences, Icahn School of Medicine at Mount Sinai, New York, United States; [11]Sema4, Stamford, United States; [12]Research and Development, Janssen Pharmaceuticals, Pennsylvania, United States; [13]Pediatric Gastroenterology and Hepatology, Department of Pediatrics, Susan and Leonard Feinstein IBD Clinical Center, Icahn School of Medicine at Mount Sinai, New York, United States

*For correspondence:
jeremiah.faith@mssm.edu

**Abstract** To identify factors that regulate gut microbiota density and the impact of varied microbiota density on health, we assayed this fundamental ecosystem property in fecal samples across mammals, human disease, and therapeutic interventions. Physiologic features of the host (carrying capacity) and the fitness of the gut microbiota shape microbiota density. Therapeutic

manipulation of microbiota density in mice altered host metabolic and immune homeostasis. In humans, gut microbiota density was reduced in Crohn's disease, ulcerative colitis, and ileal pouch-anal anastomosis. The gut microbiota in recurrent *Clostridium difficile* infection had lower density and reduced fitness that were restored by fecal microbiota transplantation. Understanding the interplay between microbiota and disease in terms of microbiota density, host carrying capacity, and microbiota fitness provide new insights into microbiome structure and microbiome targeted therapeutics.

**Editorial note:** This article has been through an editorial process in which the authors decide how to respond to the issues raised during peer review. The Reviewing Editor's assessment is that all the issues have been addressed (see decision letter).

DOI: https://doi.org/10.7554/eLife.40553.001

## Introduction

Population density is a fundamental parameter in understanding the health and function of any ecosystem, yet we know little about which host and microbial factors contribute to the density of organisms in the gut microbiota (*i.e.*, gut microbiota density). The relationships uncovered between the gut microbiota and health over the past decade have largely focused on relative differences in community composition, estimated with culture-independent 16S rRNA gene (*Caporaso et al., 2010*; *Schloss et al., 2009*) or shotgun metagenomic sequencing (*Segata et al., 2012*). The microbiome's influence on host physiology likely depends on the number – and not just the type – of bacteria interfacing with the host. Therefore, understanding factors driving gut microbiota density, as well as the impact of microbiota density on health, may advance the therapeutic potential of the microbiota.

Microbiota density has previously been measured with colony-forming units, DNA spike-ins (*Satinsky et al., 2013*; *Stämmler et al., 2016*), qPCR (*Mahowald et al., 2009*; *Rey et al., 2013*), flow cytometry (*Props et al., 2017*; *Reyes et al., 2013*; *Vandeputte et al., 2017b*), and microbial DNA quantification (microbial DNA per mass of sample) (*Faith et al., 2011*; *Llewellyn et al., 2018*; *Reyes et al., 2013*). Here, we use fecal microbial DNA content to estimate gut microbiota density, since it correlates with flow cytometry counts and colony-forming units (CFU), and it can be easily incorporated into standard microbiome sequencing workflows by weighing the sample (*Reyes et al., 2013*). We investigate host and microbial factors that contribute to microbiota density across a diverse set of mammalian microbiomes, study the impact of microbiota density on host adiposity and immune function in controlled mouse models, and describe microbiota density changes in disease and the resolution of those alterations after therapy.

## Results

### The natural variation of gut microbiota density in mammals is driven by host and microbial factors

In macroecology, carrying capacity is the maximal density of organisms supported by an ecosystem and is broadly dictated by the resources (*e.g.*, food, water, and habitat) in the environment. Whether or not the collection of species in an environment can reach the carrying capacity depends on their ability to efficiently utilize the available resources (*i.e.*, the community's fitness for the environment). To explore the contribution of host carrying capacity and gut microbiota fitness to microbiota density, we first collected fecal material from sixteen different mammalian species (*Supplementary file 1*) in order to sample a diverse range of host intestinal architectures and gut microbial community compositions. Using methods optimized to assay fecal microbiota density with greater throughput (see Materials and methods and *Figure 1—figure supplement 1*), we observed significant differences in microbiota density across the mammalian species (H = 69.0, p = $6.72 \times 10^{-9}$; Kruskal-Wallis) with a 216-fold difference between the median of the most dense and least dense gut microbiota (*Figure 1A*). At the higher taxonomic rank of order, where we sampled at least two unique species (*Atriodactyla*, *Carnivora*, *Primates*, and *Rodentia*), we still found significant differences in microbiota density (H = 39.0, p = $3.39 \times 10^{-9}$; Kruskal-Wallis), suggesting that evolutionarily conserved host features impact microbiota density. We found no correlation between microbiota density and either

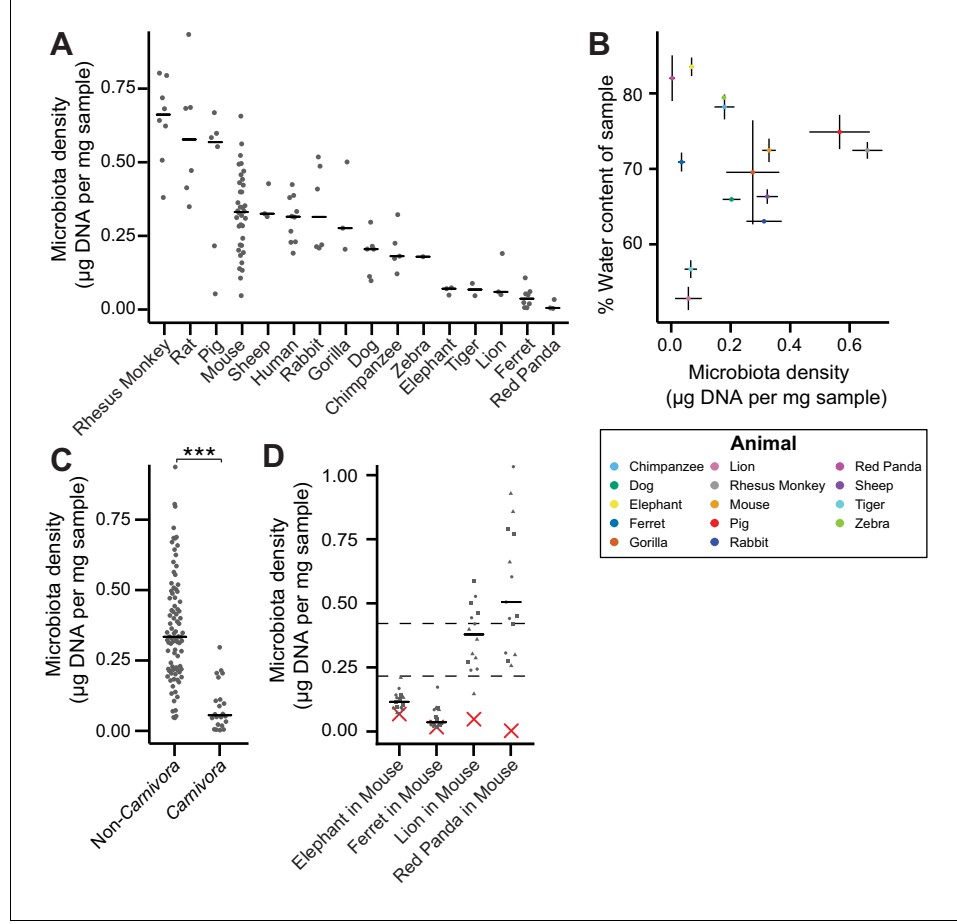

**Figure 1.** The natural variation in gut microbiota density across mammals is driven by host and microbial factors. (**A**) Fecal microbiota density varies across mammalian species. (**B**) Microbiota density and water content of fecal samples are not correlated. (**C**) Animals from the order *Carnivora* have a reduced microbiota density compared to mammals from other orders. (**D**) Different mammalian gut microbiotas transplanted into germ-free Swiss Webster mice (n = 3 per group) vary in their fitness to reach microbiota densities similar to mouse microbiotas. In (**A, C, and D**) points depict individual samples, and bars indicate median. In (**B**) points and lines indicate median values ± SEM. In (**D**) a red *X* indicates the microbiota density of the original mammalian sample, while dashed lines represent IQR of conventional Swiss Webster mice. ***p < 0.001. Source data available for (**A-D**). 16S rRNA gene amplicon sequencing data is available for (**A and D**) (see Materials and methods).

DOI: https://doi.org/10.7554/eLife.40553.002

The following source data and figure supplements are available for figure 1:

**Source data 1.** Microbiota density in mammalian samples.
DOI: https://doi.org/10.7554/eLife.40553.005
**Source data 2.** Fecal water content of mammalian samples.
DOI: https://doi.org/10.7554/eLife.40553.006
**Source data 3.** Microbiota density of gnotobiotic mice colonized with mammalian microbiome samples.
DOI: https://doi.org/10.7554/eLife.40553.007
**Figure supplement 1.** DNAse Inactivation Buffer DNA extraction method (DIB), phenol:chloroform extraction, and culture-based measurements of microbiota density yield consistent results.
DOI: https://doi.org/10.7554/eLife.40553.003
**Figure supplement 2.** Microbiota density is not correlated with body mass.
DOI: https://doi.org/10.7554/eLife.40553.004

fecal water content ($\rho$ = -0.0418, p = 0.892, Spearman; *Figure 1B*) or host size (mass) ($\rho$ = -0.364, p = 0.167, Spearman; *Figure 1—figure supplement 2*). Nonetheless, animals from order *Carnivora* (dog, ferret, lion, red panda, and tiger), with simple gut architectures adapted to carnivorous diets, had significantly reduced microbiota densities compared with the rest of the mammals studied (p = 6.14 x $10^{-10}$, Mann-Whitney, *Figure 1C*).

To assay the relative contributions of the host (*i.e.*, carrying capacity) and the microbiota (*i.e.*, microbiota fitness) to microbiota density, we utilized germ-free mice with controlled host carrying capacity (*i.e.*, fixed diet, genetics, and environment) transplanted with the microbiotas of different mammals. Although there are clear caveats to assaying properties of the microbiota in a non-native host, several prior studies have demonstrated that germ-free microbiota transplantations from other mammals can recapitulate many aspects of the microbial community (*Goodman et al., 2011*; *Ridaura et al., 2013*; *Seedorf et al., 2014*) and even host physiology (*Britton et al., 2019*; *Cekanaviciute et al., 2017*; *De Palma et al., 2017*; *Sampson et al., 2016*) in the murine host. Importantly, these microbiota transplant experiments provide an experimental tool to estimate relative differences in fitness between microbiotas because each microbiota is transplanted into one or more replicate murine hosts with the same carrying capacity. In germ-free Swiss Webster mice colonized with four of the lowest density microbiotas in our initial screen (lion, elephant, ferret, and red panda), the lion and red panda microbiotas reached higher microbiota densities in the mouse than in the native host (*Figure 1D*), suggesting their densities were limited by the carrying capacity of their host (which could include factors like intestinal architecture, host diet and host social behaviors). The elephant and ferret microbiotas colonized mice at densities comparable to those in the native host and significantly less dense than a mouse microbiota (*Figure 1D*), suggesting their densities are limited by the fitness of each microbiota that cannot reach the mouse carrying capacity. Altogether, these mammalian microbiota samples and germ-free transfer experiments demonstrate that as in macroecology, microbiota density represents the combined influence of host carrying capacity and community fitness.

## Manipulation of colonic microbiota density alters host physiology

To broadly assess the impact of therapeutics on gut microbiota density, we provided SPF mice with one of 20 orally administered drugs, including antibiotics, anti-motility agents, and laxatives (*Supplementary file 2*). Only 9 of the 14 tested antibiotics significantly decreased gut microbiota density compared to untreated animals (p < 0.05 for each; Kruskal-Wallis rank sum test, followed by a Dunn's test with Bonferroni correction), and the taxa reduced by each antibiotic were not strongly reflective of the antibiotic's spectrum (see Supplemental Results and *Figure 2—figure supplement 1*). Amongst these 9 density-reducing antibiotics, there were substantial differences in each drug's depleting capacity (*Figure 2A*). Of the laxatives, PEG 3350 reduced microbiota density (p = 2.22 x $10^{-4}$), while lactulose increased it (p = 0.0279). The anti-motility agent loperamide and the proton pump inhibitor omeprazole had no significant effect. Across the pharmacologics, we never observe high microbiota density with low alpha diversity, which drives a significant correlation between alpha diversity and microbiota density ($\rho$ = 0.628, p < 0.0001, Spearman correlation; *Figure 2—figure supplement 2H*). However, we commonly observe high alpha diversity with low microbiota density (*e.g.*, animals given metronidazole; *Figure 2—figure supplement 2H*), suggesting changes in microbiota density do not strictly correspond to changes in alpha diversity (see Supplemental Results and *Figure 2—figure supplement 2*). As with our results in the mammals, we found no correlation between microbiota density and fecal water content across the tested pharmacologics ($\rho$ = -0.338, p = 0.411, Spearman; *Figure 2—figure supplement 3F*).

Comparing antibiotic-treated or germ-free mice with conventional mice has demonstrated the influence of the microbiota on a range of physiological measures (*Atarashi et al., 2011*; *Faith et al., 2014*; *Bäckhed et al., 2004*; *Geuking et al., 2011*; *Ivanov et al., 2009*; *Mortha et al., 2014*; *Muller et al., 2014*; *Bongers et al., 2014*; *Ridaura et al., 2013*; *Wostmann and Bruckner-Kardoss, 1959*; *Zhang et al., 2015*). To better understand the impact of microbiota density on host physiology, we selected five antibiotics (ampicillin, ciprofloxacin, clindamycin, polymyxin B, vancomycin) based on their varying ability to decrease microbiota density (*Figure 2A*). As expected, treating 4-week old SPF C57BL/6J mice with each antibiotic in their drinking water for four weeks (n = 6 mice per antibiotic, 9 SPF antibiotic-free controls, and 6 germ-free controls) led to a range of density reductions across the experimental groups (1.1 – 36.0 fold; *Figure 2—figure supplement 3A*). We

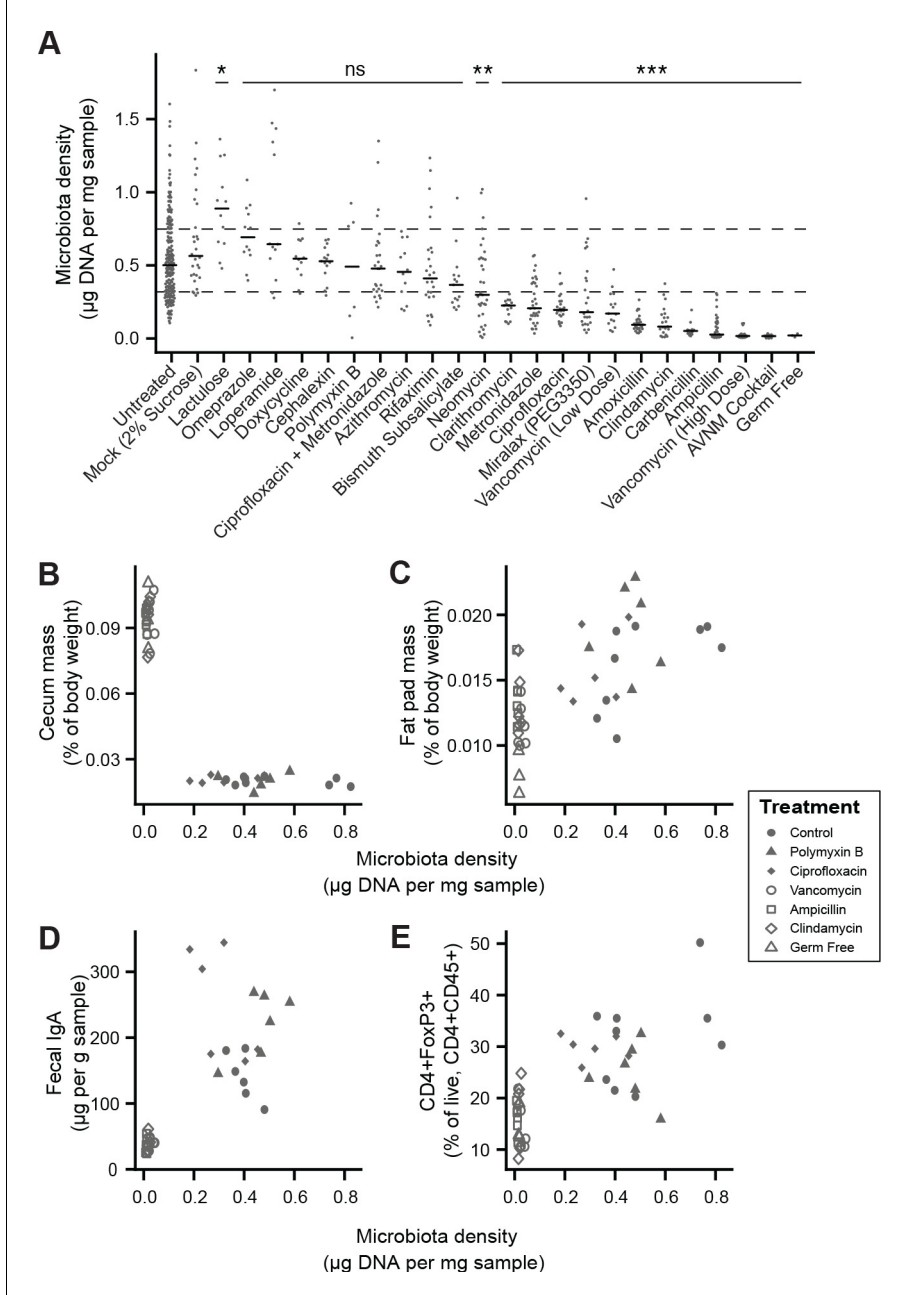

**Figure 2.** Manipulation of colonic microbiota density alters host physiology. (**A**) Pharmacologic interventions differentially alter microbiota density in SPF C57BL/6J mice. Samples from 3 to 12 (mean = 6) mice per group. (**B–E**) Antibiotic-induced changes in microbiota density significantly correlate with (**B**) host cecum size, (**C**) adiposity, (**D**) fecal IgA, and (**E**) colonic lamina propria FoxP3 +T regulatory cells. n = 6 mice per antibiotic group, 9 SPF antibiotic-free controls, and six germ-free controls. In (**A**), dashed lines represent the IQR of untreated SPF C57BL/ 6J mice and AVNM = ampicillin, vancomycin, neomycin, metronidazole. Statistical tests performed for individual treatment conditions vs untreated using Kruskal-Wallis with Dunn's post-test corrected for multiple comparisons with the Bonferonni correction. Bars indicate median. ns = not significant, *p < 0.05, **p < 0.01, and ***p < 0.001. In (**B-E**) points represent individual mice. Shapes indicate treatment group. Source data available for (**A-E**). 16S rRNA gene amplicon sequencing data is available for **A** (see Materials and methods).
DOI: https://doi.org/10.7554/eLife.40553.008

The following source data and figure supplements are available for figure 2:

**Source data 1.** Microbiota density of mice treated with pharmacologics.
DOI: https://doi.org/10.7554/eLife.40553.013

*Figure 2 continued on next page*

*Figure 2 continued*

**Source data 2.** Microbiota density and phenotypic changes in antibiotic-treated mice.
DOI: https://doi.org/10.7554/eLife.40553.014
**Source data 3.** Fecal water content of mice diets with varied fiber sources and protein content.
DOI: https://doi.org/10.7554/eLife.40553.015
**Source data 4.** Fecal water content of mice diets with varied fiber sources and protein content.
DOI: https://doi.org/10.7554/eLife.40553.016
**Figure supplement 1.** *In vivo* antibiotic spectrum of activity.
DOI: https://doi.org/10.7554/eLife.40553.009
**Figure supplement 2.** Alteration of the absolute murine fecal microbiota by pharmacologics, and the relationship between alpha diversity and microbiota density in pharmacologic interventions.
DOI: https://doi.org/10.7554/eLife.40553.010
**Figure supplement 3.** Phenotypic changes observed in antibiotic-treated mice.
DOI: https://doi.org/10.7554/eLife.40553.011
**Figure supplement 4.** Fecal water content and microbiota density can be manipulated independently by diet.
DOI: https://doi.org/10.7554/eLife.40553.012

found a significant negative correlation between cecum size and microbiota density ($\rho$ = -0.729, p = 2.46 x $10^{-7}$, Spearman; *Figure 2—figure supplement 3B*). Epididymal fat pad mass, fecal IgA, and lamina propria FoxP3$^+$CD4$^+$ regulatory T cells were each positively correlated with microbiota density ($\rho_{fat}$ = 0.587, $p_{fat}$ = 6.11 x $10^{-5}$; $\rho_{IgA}$ = 0.783, $p_{IgA}$ = 3.35 x $10^{-7}$; $\rho_{Treg}$ = 0.639, $p_{Treg}$ = 5.31 x $10^{-6}$; Spearman; *Figure 2—figure supplement 3C–3E*). The strength of these associations is independent of the water content of the feces. Using group averages, the Spearman's correlations are the same for dry and wet microbiota density vs phenotypes (*i.e.*, the rank order of density does not change when using dry weights). Furthermore, when estimating the relationships between microbiota density and host physiology with linear models we find that wet weight is a better predictor of changes in cecum size, epididymal fad pad mass, fecal IgA, and FoxP3$^+$CD4$^+$ regulatory T cells than dry weight.

## Microbiota density in inflammatory bowel disease (IBD)

To characterize the impact of host health status on gut microbiota density, we collected fecal samples from 70 healthy controls, 138 subjects with Crohn's disease (CD), 97 subjects with ulcerative colitis (UC), and 19 subjects with UC that had undergone an ileal pouch-anal anastomosis (IPAA) procedure following total colectomy. Concordant with prior work using phylum-specific qPCR (*Frank et al., 2007*) and flow cytometry (CD-only; *Vandeputte et al., 2017b*), subjects with IBD had decreased microbiota density compared to healthy controls ($p_{UC}$ = 0.00181, $p_{CD}$ = 1.77×$10^{-4}$, $p_{IPAA}$ = 2.40×$10^{-5}$, each vs Healthy, Kruskal-Wallis rank sum test, followed by a Dunn's test with Bonferroni correction; *Figure 3A*), even when excluding individuals receiving antibiotics (*Figure 3—figure supplement 1A*). Individuals with active CD, as well as IPAA subjects, had increased fecal water content compared to healthy individuals ($p_{active\ CD}$ = 0.036, $p_{IPAA}$ = 0.0184, each vs Healthy; Tukey's HSD), while individuals with UC or inactive CD did not. Nonetheless, the decrease in microbiota density in IBD compared to healthy controls was consistent across individuals with active disease or inactive disease ($p_{active\ IBD}$ = 7.68 x $10^{-5}$, $p_{active\ CD}$ = 0.000466, $p_{inactive\ IBD}$ = 0.00229, $p_{inactive\ CD}$ = 0.0479, each vs Healthy, Kruskal-Wallis rank sum test, followed by a Dunn's test with Bonferroni correction; *Figure 3B*), demonstrating that the microbiota density changes in IBD were not simply driven by the increased fecal water content that occurred with active inflammation in CD.

To associate changes in microbiota composition with the altered microbiota density in individuals with IBD, we performed 16S rRNA gene amplicon sequencing of the fecal DNA (*Figure 3C–3D*). In line with previous studies (*Frank et al., 2007*; *Gevers et al., 2014*; *Gophna et al., 2006*; *Jacobs et al., 2016*), the IBD microbiome had a decreased alpha diversity compared to healthy subjects ($p_{UC}$ = 0.00339, $p_{CD}$ = 2.39×$10^{-9}$, $p_{IPAA}$ = 1.17×$10^{-12}$, each vs Healthy; Kruskal-Wallis rank sum test, followed by a Dunn's test with Bonferroni correction; *Figure 3—figure supplement 1B*). When we multiplied each taxa's relative abundance by the microbiota density to calculate their absolute abundances, we found decreases in gut microbiota density were most significantly correlated with decreases in Firmicutes, while Proteobacteria were the only one of the four major phyla in the

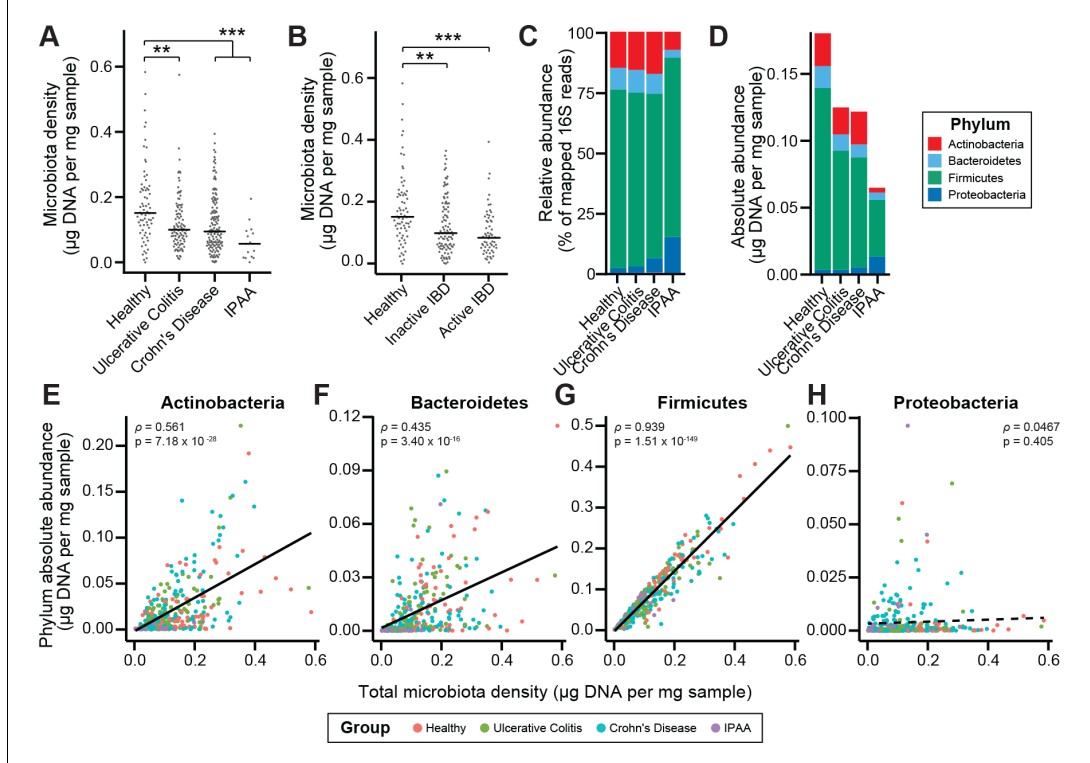

**Figure 3.** Microbiota density is altered in IBD. (**A**) Subjects with ulcerative colitis and Crohn's disease, as well as subjects who have undergone ileal pouch-anal anastomosis (IPAA) have reduced microbiota density compared to healthy controls. (**B**) The reduction in microbiota density in IBD patients is independent of disease activity. (**C–D**) 16S rRNA gene sequencing reveals phylum-level changes in (**C**) relative and (**D**) absolute abundances of the microbiota in subjects with UC, CD, and IPAA compared to healthy controls. (**E–H**) The absolute abundance of all of the major phyla are strongly correlated with microbiota density, with the exception of Proteobacteria, whose abundance is largely constant. In (**A-C**) bars indicate median, **p < 0.01, and ***p < 0.001 (Kruskal-Wallis with Dunn's post-test corrected for multiple comparisons with the Bonferonni correction). In (**C**) each point represents the average microbiota density for an individual mouse before or after the initiation and development of colitis. In (**E-H**) points represent individual subjects and colors indicate their health status. Source data available for (**A and B**). 16S rRNA gene amplicon sequencing data is available for (**C-H**) (see Materials and methods).

DOI: https://doi.org/10.7554/eLife.40553.017

The following source data and figure supplement are available for figure 3:

**Source data 1.** Microbiota density and diversity in individuals with IBD or IPAA.
DOI: https://doi.org/10.7554/eLife.40553.019

**Figure supplement 1.** The microbiota of IBD and IPAA subjects.
DOI: https://doi.org/10.7554/eLife.40553.018

gut microbiota that were not correlated with microbiota density (*Figure 3E–3H*). These results from measuring the density of each phyla provide a novel insight compared to previous studies that associated a relative increase in the proportion of Proteobacteria with IBD (*Frank et al., 2007*; *Gevers et al., 2014*). We show here that in absolute terms, Proteobacteria are able to sustain a constant density in the individuals with IBD while the remaining phyla decrease in density.

## Fecal microbiota transplants restore microbiota density and microbiota fitness

Given the large difference in the microbiota between healthy individuals and those with recurrent *Clostridium difficile* infection (rCDI) (*Figure 4—figure supplement 1A and B*; *Seekatz et al., 2014*; *Shankar et al., 2014*), we hypothesized that on a mechanistic level, FMT bolsters colonization resistance by improving gut microbiota fitness. In fecal samples from FMT donors and their rCDI FMT recipients prior to and after FMT, we observed that the rCDI gut microbiota has a significantly lower microbiota density than the donor microbiota, and that FMT increased microbiota density (p < 0.05

for all comparisons, Kruskal-Wallis rank sum test, followed by a Dunn's test with Bonferroni correction; *Figure 4A*). We did not observe any differences in fecal water content between the donors and recipients before or after FMT (p > 0.2 for all comparisons, Tukey's HSD). In addition, we found that rCDI FMT recipients had both a relative and absolute increase in Proteobacteria that was significantly reduced by FMT (*Figure 4B and C*, and *Figure 3—figure supplement 1C–1F*). These data suggest that FMT restores higher densities of Bacteroidetes, Firmicutes, and Actinobacteria to more fully realize the host's carrying capacity. However, these results may be confounded by the fact that the individuals with rCDI have been exposed to antibiotic treatment prior to their FMT, and as we showed in *Figure 2A*, antibiotics may reduce microbiota density.

To separate the host physiologic and pharmacologic factors that might impact our understanding of community fitness in rCDI, we utilized a gnotobiotic murine model of FMT (*Figure 4D*) where

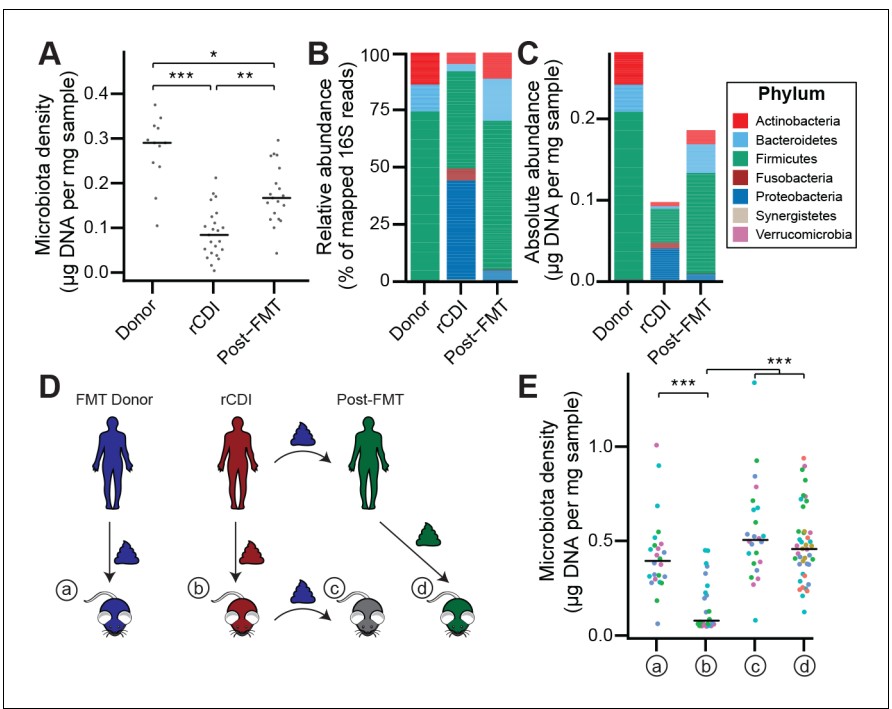

**Figure 4.** The rCDI microbiota has a fitness defect that is therapeutically treatable by FMT. (**A**) rCDI subjects have reduced microbiota densities that are significantly increased upon FMT with donor microbiotas. (**B and C**) Following FMT, the composition of the microbiota of individuals with rCDI is restored to more closely resemble that of healthy donors in both (**B**) relative and (**C**) absolute terms. (**D**) Germ-free mice were colonized with the microbiota from FMT Donors (**a**) or individuals with rCDI that underwent FMT (**b**). These mice then received the microbiota from the FMT donor corresponding to the clinical FMT (**c**) which could be compared to germ-free mice colonized with the Post-FMT sample from the individual who received the FMT (**d**). (**E**) Microbiota density in mice from the experimental scheme described in (**D**) showed decrease in microbiota fitness prior to FMT and an increase in microbiota density following FMT demonstrating the restoration of community fitness. In (**A and E**) points represent individual samples, bars indicate median, *p < 0.05, **p < 0.01, and ***p < 0.001 (Kruskal-Wallis with Dunn's post-test corrected for multiple comparisons with the Bonferonni correction). In (**E**), colors represent each one of five different FMT donor-recipient pairs. Source data available for **A** and **E**. 16S rRNA gene amplicon sequencing data is available for **B**, **C**, and **E** (see Materials and methods).
DOI: https://doi.org/10.7554/eLife.40553.020

The following source data and figure supplement are available for figure 4:

**Source data 1.** Microbiota density of FMT recipients and donors.
DOI: https://doi.org/10.7554/eLife.40553.022

**Source data 2.** Microbiota density of gnotobiotic mouse model of FMT.
DOI: https://doi.org/10.7554/eLife.40553.023

**Figure supplement 1.** FMT changes the microbiome of individuals with rCDI to resemble that of healthy donors.
DOI: https://doi.org/10.7554/eLife.40553.021

germ-free mice were initially colonized with the fecal material of individuals with rCDI for 3 weeks prior to a single transplant of fecal material via oral gavage from a second human donor – the same healthy FMT donor used for the transplant clinically. The ex-germ-free mice therefore model the fecal microbiota transplant but in a fixed environment, with a controlled diet, and no antibiotic confounder. As a control, we colonized germ-free mice with the FMT donor microbiota alone (*Figure 4D*). The microbiota density of mice colonized with the healthy samples (a) was greater than that of mice colonized with rCDI samples (b) (p = $1.79 \times 10^{-4}$, Kruskal-Wallis rank sum test, followed by a Dunn's test with Bonferroni correction; *Figure 4E*), suggesting that rCDI individuals have a reduced microbiota fitness compared to healthy donors. Following the introduction of the healthy donor microbiota to the mice colonized with the rCDI microbiota (c), we observed increased microbiota density in these mice (p = $6.88 \times 10^{-8}$, Kruskal-Wallis rank sum test, followed by a Dunn's test with Bonferroni correction; *Figure 4E*), implying a restoration of microbiota fitness. Furthermore, when we colonize germ-free mice with the microbiota of the individuals with rCDI 6–12 months after they received an FMT (d), we find that their microbiota fitness had been restored, just as in our mouse FMT model (p = $8.60 \times 10^{-8}$, Kruskal-Wallis rank sum test, followed by a Dunn's test with Bonferroni correction; *Figure 4E*). These findings in the mice model recapitulate the data in our human cohort of FMT recipients and suggest that FMT successfully treats the fitness defect of the rCDI community.

## Discussion

The DNA-based microbiota density estimation method employed here and in previous studies (*Faith et al., 2011*; *Llewellyn et al., 2018*; *Reyes et al., 2013*) has the advantage that it can be incorporated into existing 16S rRNA and metagenomic workflows by simply weighing the input sample and ensuring the input mass of fecal material is within the linear range of the DNA extraction protocol. Incorporating microbiota density into standard culture independent microbiome workflows would greatly broaden our understanding of factors that drive one of the most fundamental properties of any ecosystem – its population density – and it would allow the broader study of absolute taxon abundances. Recent work has demonstrated that the amount of live/dead bacteria can vary between fecal samples (*Costea et al., 2017*; *Maurice et al., 2013*; *Sinha et al., 2017*), which would not be captured by a DNA-based density metric. However, in practice we found that the influence of any variation from live/dead bacteria was sufficiently low that it did not influence the major conclusions of this study; we observed a very significant correlation between the viability-based CFU density measurement and the DNA-based one and all of the major relationships observed in this study were consistent across both approaches (*i.e.*, variation across mammals, IBD and IPAA lower density than healthy, rCDI lower density than FMT donor or rCDI post-transplant; *Figure 1—figure supplement 1D*).

Although bacteria dominate the gut microbiota and were the primary focus of this study, this approach could also be used to account for the non-bacterial (*e.g.*, fungal, viral, protozoan) components of the microbiome, assuming an appropriate DNA extraction method was chosen to efficiently lyse these broader microbial groups. While recent studies that examined the fungal microbiome have shown that fungi can alter the efficacy of FMT in rCDI for selected individuals (*Zuo et al., 2018*), we did not detect a substantial contribution of fungi in the metagenome of our rCDI samples extracted with phenol:chloroform and bead beating (see Supplemental Results). As described above, microbial density is highly correlated with CFU and ultimately each microbial genome is associated with a single organism. Therefore, as long as the relative abundance measure is scaled by the size of each organism's genome (i.e., genome equivalents) the density of each species estimated by this DNA-based method should also roughly reflect the CFU density of each species with the same caveats of 16S rRNA copy number and genome copy number that apply to all microbiome studies. As measures of microbiota density seek to become more accurate, well-curated databases that contain information on fundamental characteristics of microbes such as cell volume or mass could provide further refinement.

Previous work has demonstrated that changes in fecal water are associated with but not necessarily causally influencing differences in microbiota composition across the human population (*Falony et al., 2016*; *Vandeputte et al., 2017a*). While both microbiota density and fecal water content vary across mammals and can be altered by pharmacologics (*Figures 1* and *2*, and *Figure 2—*

figure supplement 3F), dietary components (*Figure 2—figure supplement 1*; *Llewellyn et al., 2018*), and host disease status (*Figures 3* and *4*), we find microbiota density is consistently not correlated with water content. In the context of altering host physiology through antibiotic manipulation of microbiota density, the best predictor of the impact of changes of microbiota density on host physiology was when density was calculated with stool wet weight, suggesting both wet and dry components of stool are important diluents in determining microbiota density and its impact on the host. Our results suggest that microbiota density may change independently of water content, implying that the density of microbes can be altered independently of water and other contents of the stool bulk, such as undigested dietary components or host tissue.

Differences in microbiota density can be influenced by both the host's carrying capacity and the fitness of the microbiota to reach the carrying capacity of a given host. We found the density of gut microbes varies across mammals and is more similar in more phylogenetically related species. Across mammals, gut architecture appears to be a major driver of density, as the lowest densities were observed in order *Carnivora*, whose short, simple intestines have a lower carrying capacity and are maladapted for microbial fermentation at high densities. The low microbiota density of the red panda, a member of *Carnivora* with a herbivorous diet, further supports intestinal architecture as a major determinant of host carrying capacity and thus a driver of microbiota density. Finally, the significantly reduced microbiota density in humans with IPAA uniquely demonstrates that changing gut architecture within a species (in this case by surgery to treat ulcerative colitis) is equally capable of influencing host carrying capacity. Outside of animal models, it is possible that other host features that may be more readily altered, such as dietary habits or social behaviors, may also influence the host carrying capacity.

Within a murine host with controlled carrying capacity (*i.e.*, fixed diet, genetics, housing, etc.), we found microbiota density can be altered with pharmacologics, with downstream consequences to host adiposity and immune function. Different antibiotics were highly varied in their ability to impact microbiota density, which could explain the mixed efficacy of antibiotics in microbiota-targeted clinical trials for complex disease and varied responses to antibiotics in animal models. Identifying more effective microbiota depleting cocktails would improve the design of such studies, while measuring microbiota density in trials with antibiotics could better stratify clinical response. Previous studies have observed that microbiota density can be manipulated by dietary changes (*Llewellyn et al., 2018*; *Sonnenburg et al., 2016*). Furthermore, we found that altering microbiota density with either diet or antibiotics could modify colitis severity (*Llewellyn et al., 2018*). Understanding the long-term effect of high or low microbiota density on health could help refine the use of diet and the microbiota in disease treatment and prevention.

We also observed that microbiota density is reduced in individuals with IBD. Coupled with our findings that changes in microbiota density can alter host metabolism and immune populations, these results suggest that chronically low microbiota density may play a role in the development or progression of disease. It might even be possible that an initial reduction in microbiota density contributes to a pro-inflammatory host immune system that creates a positive feedback loop that sustains a low microbiota density. It is also possible that a low microbiota density, if due to low microbiota fitness, has reduced colonization resistance, allowing for pathogens or pathobionts to take hold and contribute to disease processes in the host (*Battaglioli et al., 2018*).

Finally, we found that the reduced microbiota density in rCDI, due to a lack of microbiota fitness, was 'druggable' by FMT. Moving forward, studying the factors that determine both host carrying capacity and microbiota fitness may allow us to predict which disease states may benefit from therapeutics that target the host versus ones that target the microbiota. By identifying components of the microbiota that confer increased fitness, we can improve our understanding of the ecological rules that govern the microbiome. For example, exploring how FMT is able to increase microbiota fitness and therefore microbiota density should provide mechanistic insights into FMT for rCDI that can be used for other potential indications for FMT. These results also suggest that routine monitoring could identify individuals with microbiota fitness deficiencies that might benefit from prophylactic microbial therapeutics to boost colonization resistance to treat or prevent disease (*Battaglioli et al., 2018*).

# Materials and methods

**Key resources table**

| Reagent type (species) or resource | Designation | Source or reference | Identifiers | Additional information |
|---|---|---|---|---|
| Antibody | Anti-Mouse/ Rat Foxp3 PE | Thermo Fisher Scientific | Cat# 12-5773-82; RRID:AB_465936 | (1:100) |
| Antibody | APC Anti-Mouse CD4 | BioLegend | Cat# 100411; RRID:AB_312696 | (1:200) |
| Antibody | APC/Cy7 Anti-Mouse CD45 | BioLegend | Cat# 103115; RRID:AB_312980 | (1:100) |
| Antibody | Goat Anti-Mouse IgA-HRP | Sigma-Aldrich | Cat# A4789; RRID:AB_258201 | (1:2000) |
| Antibody | Goat Anti-Mouse IgA-UNLB | SouthernBiotech | Cat# 1040–01; RRID:AB_2314669 | Working concentration 1 ng/μL |
| Chemical compound, drug | Amoxicillin | Sigma-Aldrich | Cat# A8523 | |
| Chemical compound, drug | Ampicillin | Sigma-Aldrich | Cat# A9518 | |
| Chemical compound, drug | Azithromycin | AK Scientific | Cat# SYN3010 | |
| Chemical compound, drug | Carbenicillin | Sigma-Aldrich | Cat# C1389 | |
| Chemical compound, drug | Cephalexin | Sigma-Aldrich | Cat# C4895 | |
| Chemical compound, drug | Ciprofloxacin | Sigma-Aldrich | Cat# 17850 | |
| Chemical compound, drug | Clarithromycin | Sigma-Aldrich | Cat# C9742 | |
| Chemical compound, drug | Clindamycin | Sigma-Aldrich | Cat# C5269 | |
| Chemical compound, drug | Doxycycline | Sigma-Aldrich | Cat# D9891 | |
| Chemical compound, drug | Lactulose | Sigma-Aldrich | Cat# 61360 | |
| Chemical compound, drug | Loperamide | Sigma-Aldrich | Cat# L4762 | |
| Chemical compound, drug | Metronidazole | Research Products International | Cat# M81000 | |
| Chemical compound, drug | Neomycin | Sigma-Aldrich | Cat# N6386 | |
| Chemical compound, drug | Omeprazole | Sigma-Aldrich | Cat# O104 | |
| Chemical compound, drug | Peroxidase Solution B | KPL | Cat# 50-65-02 | |
| Chemical compound, drug | PhenoL:Chloroform :IAA, 25:24:1, pH 6.6 | Thermo Fisher Scientific | Cat# AM9732 | |
| Chemical compound, drug | PM Buffer | Qiagen | Cat# 19083 | |
| Chemical compound, drug | Polyethylene Glycol 3350 | Miralax | Product # 11523–723 | |
| Chemical compound, drug | Polymyxin B | Sigma-Aldrich | Cat# P0972 | |
| Chemical compound, drug | Rifaximin | Sigma-Aldrich | Cat# R9904 | |

*Continued on next page*

*Continued*

| Reagent type (species) or resource | Designation | Source or reference | Identifiers | Additional information |
|---|---|---|---|---|
| Chemical compound, drug | RNAlater Stabilization Reagent | Qiagen | Cat# 76104 | |
| Chemical compound, drug | Sodium dodecyl sulfate (SDS) | Sigma-Aldrich | Cat# 75746 | |
| Chemical compound, drug | TMB Peroxidase Substrate | KPL | Cat# 50-76-02 | |
| Chemical compound, drug | Vancomycin | Amresco | Cat# 990 | |
| Commercial assay or kit | Bioanalyzer 6000 Nano Kit | Agilent | Cat 5067–1511 | |
| Commercial assay or kit | Foxp3 Fixation/ Permeabilization Buffer Set | BioLegend | Cat# 421403 | |
| Commercial assay or kit | NEBNext Ultra II DNA Library Prep Kit | New England BioLabs | Cat# E7645L | |
| Commercial assay or kit | QIAquick 96 PCR Purification Kit | Qiagen | Cat# 28181 | |
| Commercial assay or kit | Quant-IT dsDNA Assay Kit – Broad Range | Thermo Fisher Scientific | Cat# Q32853 | |
| Commercial assay or kit | Quant-IT dsDNA Assay Kit – High Sensitivity | Thermo Fisher Scientific | Cat# Q33130 | |
| Commercial assay or kit | RNeasy Mini Kit | Qiagen | Cat# 74104 | |
| Commercial assay or kit | Zombie Aqua Fixable Viability Kit | BioLegend | Cat# 423101 | |
| Other | 0.1 mm diameter zirconia /silica beads | BioSpec | Cat# 11079101z | |
| Other | 1.0 mL collection tubes | Thermo Fisher Scientific | Cat# 3740 | |
| Other | 2.0 mL collection tubes | Axygen | Cat# SCT-200-SS-C-S | |
| Other | Agencourt AMPure XP Beads | Beckman Coulter | Cat# A63880 | |
| Other | Bioruptor Pico | Diagenode | Cat# B01060010 | |
| Other | Collagenase VIII | Sigma-Aldrich | Cat# C2139 | |
| Other | DNase1 | Sigma-Aldrich | Cat# DN25 | |
| Other | LSR II Flow Cytometer | BD Biosciences | SORP | |
| Other | Mini-Bead beater-96 | BioSpec | Cat# 1001 | |
| Other | NEBNext Ultra Q5 Master Mix | New England BioLabs | Cat# M0544L | |
| Other | SPRIselect Beads | Beckman Coulter | Cat# B23317 | |

*Continued on next page*

*Continued*

| Reagent type (species) or resource | Designation | Source or reference | Identifiers | Additional information |
|---|---|---|---|---|
| Other | Synergy HTX Multi-Mode Microplate Reader | BioTek | http://www.biotek.com | |
| Other, deposited data | Greengenes reference database version 13_8 | *DeSantis et al., 2006* | http://greengenes.lbl.gov | |
| Other, deposited data | Microbiota 16S rDNA gene sequences | This paper | SRA Project #: PRJNA413199 | |
| Other, deposited data | Mus musculus mm10 genome | UCSC | http://genome.ucsc.edu | |
| Other, deposited data | Shotgun metagenomic sequencing data | This paper | SRA Project #: PRJNA413199 | |
| Sequence-based reagent (primers) | 16S V4 (515–806) F 5'-GTGCCAGCA GCCGCGGTAA-3' | IDT (*Relman et al., 1992*) | N/A | |
| Sequence-based reagent (primers) | 16S V4 (515–806) R 5'-GGACTACCA GGGTATCTAAT-3' | IDT (*Relman et al., 1992*) | N/A | |
| Sequence-based reagent (primers) | Mouse TNFa (6455–6718) F 5'-GGCTTTCCG AATTCACTGGAG-3' | IDT (*Nitsche et al., 2001*) | N/A | |
| Sequence-based reagent (primers) | Mouse TNFa (6455–6718) R 5'-CCCCGGCC TTCCAAATAAA-3' | IDT (*Nitsche et al., 2001*) | N/A | |
| Software, algorithm | FACSDiva | BD Biosciences | http://www.bdbiosciences.com/us/instruments/research/software/flow-cytometry-acquisition/bd-facsdiva-software/m/111112/overview | |
| Software, algorithm | FLASH | *Magoč and Salzberg, 2011* | http://ccb.jhu.edu/software/FLASH/ | |
| Software, algorithm | FlowJo (version 10) | Treestar | https://www.flowjo.com/solutions/flowjo/downloads | |
| Software, algorithm | MetaPhlAn2 | *Truong et al., 2015* | N/A | |
| Software, algorithm | *Multcomp R package* | *Hothorn et al., 2008* | https://cran.r-project.org/package=multcomp | |
| Software, algorithm | *Phyloseq R package* | *McMurdie and Holmes (2013)* | https://joey711.github.io/phyloseq | |
| Software, algorithm | QIIME (version 1.9.1) | *Caporaso et al., 2010* | http://qiime.org | |
| Software, algorithm | R | *R Core Team, 2017* | https://www.R-project.org | |
| Strain, strain background (mus musculus) | C57BL/6J mice | Jackson Laboratory | Stock #000664 | |
| Strain, strain background (mus musculus) | Swiss Webster mice | Taconic Biosciences | SW-M and SW-F | |

## Mammalian samples

Fecal samples from the mammals used in this study were collected either from laboratory animals housed and maintained at the Icahn School of Medicine at Mount Sinai (New York, NY), or from animals at the Zoo Knoxville (Knoxville, TN). Approximate animal masses were curated from the literature (*Blandford, 1987*; *Lambert, 1998*; *Ebinger, 1974*; *Garland, 1983*; *Roberts and Gittleman, 1984*; *Smith and Jungers, 1997*).

## Mice

Specific pathogen free (SPF) mice were purchased from Jackson Labs (C57BL/6J) or Taconic (Swiss Webster Mice). Germ-free (GF) WT C57BL/6J (Jackson), and Swiss Webster (Taconic) mice were housed in standard, commercially available flexible film isolators. To generate gnotobiotic mice from human or mammalian fecal samples, GF mice were gavaged with 200 μL of clarified stool from the source. Four week old male mice were used for the antibiotic treatment phenotyping experiments (Figure 3). All other experiments used both male and female mice between 4 and 6 weeks old. Swiss Webster mice were used to perform gnotobiotic experiments. All animal experiments in this study were approved by Institutional Animal Care and Use Committee (IACUC) of the Icahn School of Medicine (protocol: IACUC-2013-1385) and were performed in accordance with the approved guidelines for animal experimentation at the Icahn School of Medicine at Mount Sinai.

## Human subjects

Individual ages 18 and over were recruited to be part of the study using a protocol approved by the Mount Sinai Institutional Review Board (HS# 11-01669). Once the coordinators went over the consent form and subjects consented to be part of the study to be published with subjects deidentified, they were given a study identification number that all their study samples were labeled with. All study samples were processed with no identifiers linked to them other than their study id. To study the microbiota of individuals with IBD, we collected fecal samples from 70 healthy controls (42 female, 28 male), with an average age of 55.1 (range: 23-73), 109 individuals with ulcerative colitis (67 female, 42 male), with an average age of 52.8 (range: 22-80), and 144 individuals with Crohn's Disease (72 female, 72 male), with an average age of 41.7 (range: 22-79). For subjects with ulcerative colitis we defined disease activity using the Mayo Endoscopic Subscore (Mayo). Individuals with a Mayo = 3 were categorized as having active disease, and individuals with a Mayo = 0 were categorized as having inactive disease. For individuals with Crohn's disease, active disease was defined as a Simple Endoscopic Score for Crohn Disease (SES-CD) $\geq$5, and inactive disease as SES-CD = 0. The remaining samples were excluded from these analyses. Stool samples were also collected from individuals with ulcerative colitis that had undergone an ileal pouch-anal anastomosis procedure following total colectomy (3 female, 12 male), with an average age of 42.93 (range: 19-68). These samples were collected from individuals in accordance with the IRB at the Tel Aviv Sourasky Medical Center. All individuals signed an informed consent. For the analysis of the change in the microbiota in recurrent *Clostridium difficile* infection following fecal microbiota transplantation, we collected samples from 11 healthy donors (8 female, 3 male; average age: 47.9, range: 25-75), 12 recipients who also had IBD (8 female, 4 male; average age: 55.3, range: 32-78), and 11 recipients who did not have IBD (9 female, 3 male; average age: 62, range: 36-87), as described in *Hirten et al. (2018)*. The study was approved by the Mount Sinai IRB.

## Fecal sample collection and pre-processing

To quantify the mass of each fecal sample or fecal sample aliquot, we pre-weighed tubes prior to sample collection and post-weighed the tubes after adding the fecal material. For mouse samples, fresh fecal samples were collected directly into the collection tubes and stored at −80°C. For all other mammalian species with larger fecal sample sizes, samples were aliquoted on dry ice or liquid nitrogen and stored at −80°C. Sample aliquot sizes were targeted in the linear range of the fecal DNA extraction protocol (approx. 50 mg in mice and <200 mg in humans) to enable quantitative yields of DNA from the fecal material. Samples weighing less than 5 mg were excluded from analysis.

## Phenol:chloroform DNA extraction

Fecal samples processed with the phenol:chloroform DNA extraction method were collected into 2.0 mL collection tubes (Axygen, SCT-200-SS-C-S). Similar to previous studies (*Reyes et al., 2013*), samples were suspended in a solution containing 282 µL of extraction buffer (20 mM Tris (pH 8.0), 200 mM NaCl, 2mM EDTA), 200 µL 20% SDS, 550 µL phenol:chloroform:isoamyl alcohol (25:24:1, pH 7.9), and 400 µL of 0.1 mm diameter zirconia/silica beads (BioSpec, 11079101z). Samples were then lysed by mechanical disruption with a Mini-Beadbeater-96 (BioSpec, 1001) for 5 minutes at room temperature. Samples were centrifuged at 4000rpm for 5 minutes to separate aqueous and organic phases. The aqueous phase was collected and mixed with 650 µL of PM Buffer (Qiagen, 19083). DNA extracts were then purified using a Qiagen PCR Purification kit (Qiagen, 28181), and eluted into 100 µL of EB buffer. Purified DNA was quantified using the Broad Range or High Sensitivity Quant-IT dsDNA Assay kit (Thermo Fisher, Q32853 and Q33130) in combination with a BioTek Synergy HTX Multi-Mode Reader.

## DNase inactivation buffer DNA extraction

Phenol:chloroform based DNA extraction with bead beating is an effective method to isolate microbial DNA from feces. However, automation of phenol:chloroform requires liquid handling robotics in an environment compatible with this hazardous chemical mixture. In addition, the variable volume of the aqueous phase produced with this method presents an obstacle for its automation. We therefore tested the DIB bead beating extraction protocol as an alternative, since by eliminating the hazardous chemicals the protocol is compatible with more high-throughput liquid handling robotics platforms.

Samples processed with the DNase Inactivation Buffer (DIB) DNA extraction method were collected into 1.0 mL tubes (Thermo Fisher, 3740). Samples were suspended in a solution containing 700 µL of DIB (0.5% SDS, 0.5 mM EDTA, 20 mM Tris (pH 8.0)) and 200 µL of 0.1 mm diameter zirconia/silica beads. Samples were then lysed by mechanical disruption and centrifuged as above. Since there is no phase separation with this method, it is straightforward to subsample the supernatant to improve the dynamic range of DNA quantification by avoiding saturating the column with DNA quantities above the binding capacity. 50-200 µL of the supernatant was transferred into new collection tubes. Depending on the volume collected, an additional volume of DIB was added in order to reach a total volume of 200 µL. Next, this DIB lysate was combined with 600 µL of PM Buffer, purified with a Qiagen PCR Purification kit, and eluted into 100 µL of EB buffer. Purified DNA was quantified using the Broad Range or High Sensitivity Quant-IT dsDNA Assay kit in combination with a BioTek Synergy HTX Multi-Mode Reader.

## 16S rRNA sequencing

DNA templates were normalized to 2 ng/µL, and the V4 variable region of the 16S rRNA gene was amplified by PCR using indexed primers as previously described (*Faith et al., 2013*). The uniquely indexed 16S rRNA V4 amplicons were pooled and purified with AMPure XP beads (Beckman Coulter, A63880) with a ratio of 1:1 beads to PCR reaction. Correct amplicon size and the absence of primer dimers were verified by gel electrophoresis. The pooled samples were sequenced with an Illumina MiSeq (paired-end 250bp). Raw sequencing files (fastq) for all 16S sequencing samples are stored in the public Sequence Read Archive (SRA) under project number PRJNA413199.

## Shotgun metagenomic sequencing

Metagenomic libraries were prepared using the NEBNext Ultra II DNA Library Prep kit (New England BioLabs, E7645L). Briefly, DNA samples were first sheared by sonication with a Diagenode Bioruptor Pico sonicator (Diagenode, B01060010) for a total of 14 cycles of 20 seconds. End repair and adapter ligation was performed as per the manufacturer's instructions. The ligation products were then purified using a double size selection with SPRIselect beads (Beckman Coulter, B23317) to retain products of 500–600 base pairs. Enrichment PCR was performed with NEBNext Ultra Q5 Master Mix (New England BioLabs, M0544L). Samples were quantified using the High Sensitivity Quant-IT dsDNA assay kit in combination with a BioTek Synergy HTX Multi-Mode Reader, checked for appropriate size by gel electrophoresis, and pooled in even proportions. The pooled libraries were then purified with double size selection using 0.6x followed by 0.2x of AMPure XP beads (Beckman Coulter, A63880). Samples were sequenced with an Illumina HiSeq (paired-end 150 bp). For

MetaPhlAn2 analysis, paired end sequence files were combined into one file per sample by concatenation of the two read files. Sequence data files (fastq) for all metagenomic sequencing samples are stored in the public Sequence Read Archive (SRA) under project number PRJNA413199.

## Fecal sample water content

Samples were collected into pre-weighed 2.0 mL collection tubes (Axygen, SCT-200-SS-C-S). After collecting a fecal sample, sample mass was determined by post-weighing the tube. To measure the water content of a sample, tubes were placed at 105°C for 24 hr, and weighed again (*Hinnant and Kothmann, 1988*). The water content of a sample was calculated as the difference in final and initial mass of the sample, divided by the initial mass.

## Pharmacologic treatment of mice

Antibiotics (and other compounds) were provided *ad libitum* to mice in their drinking water, when possible. All of the pharmacologics were prepared into a 2% sucrose solution (which also served as the control treatment) and sterilized with a 0.22 µm filter. Compounds that were not readily water-soluble were administered to mice via oral gavage of 200 µL once per day, as indicated in *Supplementary file 1*. Unless identified otherwise, antibiotic and pharmacologic concentrations were calculated using a maximal clinical dose (taken from the online clinical resource UpToDate.com) or from previous studies (*Atarashi et al., 2011*; *Kashyap et al., 2013*; *Larsson et al., 1983*; *Vaishnava et al., 2011*; *Bryant et al., 1988*), assuming a 20 g mouse that drinks 3 mL water per day.

## Measurement of fecal immunoglobulin A

Fecal pellets were collected and massed. To each fecal pellet, 1 mL of sterile PBS was added per 100 mg feces. Each sample was homogenized without beads in a Mini-Beadbeater-96 for 3 min (Bio-Spec, 1001) followed by vortexing for 3 min. Samples were centrifuged at 9000g for 10 min at 4°C and supernatants were collected. Immunoglobulin A was measured by ELISA. Plates were coated with a working concentration of 1 ng/µL of goat anti-mouse IgA-UNLB (SouthernBiotech Cat# 1040-01, RRID:AB_2314669), and then blocked with 1% BSA in PBS overnight at 4°C. Wells were washed with washing buffer (0.1% Tween-20 in PBS) 3 times. Then, fecal supernatant was diluted in dilution buffer (0.1% Tween-20, 1% BSA in PBS), added to each well, and incubated overnight at 4°C. The wells were washed again with washing buffer 5 times, and incubated for 2 hours at room temperature with a 1/2000 dilution of goat anti-mouse IgA-HRP (Sigma-Aldrich Cat# A4789, RRID: AB_258201) in dilution buffer. Following the incubation, the wells were washed 5 times with PBS/Tween-20. Next, TMB substrate was added to wells for 1 minute (KBL, 50-76-02 and 50-65-02), and the reaction was quenched using 1M $H_2SO_4$. Absorbance at 450 nm was measured using a BioTek Synergy HTX Multi-Mode Reader. Samples were quantified against a standard curve from 1000 ng/mL to 0.5 ng/mL.

## CFU assay

We performed colony forming unit assays to obtain a culture-dependent measurement of microbiota density that also incorporates viability, as only live microbes will form colonies in this assay. Fecal samples were stored at −80°C after sampling. Prior to plating larger samples were pulverized under liquid nitrogen. Approximately 500 mg of fecal sample was homogenized in 12 ml of rich broth and filtered with a 100 µM filter to remove particulate matter (*Britton et al., 2019*). Serial dilutions of this clarified fecal slurry were plated on chocolate agar and grown in an anaerobic chamber at 37°C for 72 hours, whereupon colonies were manually quantified and normalized to CFU/g feces.

## Colonic lamina propria immune populations

Colonic lamina propria immune cell populations were measured as previously described (*Britton et al., 2019*). Briefly, colonic tissue was dissected and placed into RPMI medium at 4°C. Tissues were then transferred into HBSS and vortexed briefly, before being transferred into dissociation buffer (10% FBS, 5 mM EDTA, 15 mM HEPES in HBSS) and shaken for 30 minutes at 110 rpm at 37°C. Tissues were washed in HBSS before digestion in HBSS containing 2% FBS, 0.5 mg/mL Collagenase VIII (Sigma C2139) and 0.5 mg/mL DNase 1 (Sigma DN25) for 30 minutes at 110 rpm at

37°C. Digested tissue was then passed through a 100 μm filter into cold RPMI medium. Samples were then centrifuged at 1500 rpm, 4°C for 5 minutes. The supernatant was removed and cells were washed once more in PBS before staining for flow cytometry. No enrichment of mononuclear cells by density centrifugation was performed. Cells were initially blocked with Fc Block (BioLegend Cat# 101320, RRID:AB_1574975) and subsequently stained for: viability (BioLegend Cat# 423101) and immunolabelled for expression of CD4 (1:200, BioLegend Cat# 100411, RRID:AB_312696) and CD45 (1:100, BioLegend Cat# 103115, RRID:AB_312980), and FoxP3 (1:100, Thermo Fisher Scientific Cat# 12-5773-82, RRID:AB_465936). Surface markers were stained before fixation and intracellular markers were stained after fixation with the FoxP3 Fixation/Permeabilization Kit (eBioscience). Samples were run on a BD LSRII and analyzed with FlowJo.

## Microbiota density and absolute abundances

We define microbiota density as the total DNA extracted from each sample (in μg) per mg of fresh sample. For samples processed with the DIB-based extraction method, the total DNA extracted is adjusted by the fraction of the supernatant that was subsampled in the DNA extraction (*e.g.,* a 100 μL subsample is 1/7th of the total volume; total sample DNA is [DNA eluted] * 7). We then are able to utilize this measurement of microbiota density to compute the absolute abundance of microbial taxa by scaling the relative abundances of microbes in a sample by the microbiota density of that sample.

## 16S rRNA gene amplicon sequencing data analysis

Paired end reads were joined into a single DNA sequence using the FLASH algorithm (*Magoč and Salzberg, 2011*). We split our pooled sequencing library by index using QIIME v 1.9.1 (*Caporaso et al., 2010*), and picked OTUs against the greengenes reference database 13_8 at 97% sequence identity (*DeSantis et al., 2006*; *McDonald et al., 2012*). The resulting OTU tables were subsequently analyzed in R (*R Core Team, 2017*) with the help of the *phyloseq* package (*McMurdie and Holmes, 2013*), and custom functions developed to convert relative abundances into absolute abundances using microbiota density data.

## Shotgun metagenomic sequencing data analysis

The metagenomic sequencing data was analyzed using MetaPhlAn2 (*Truong et al., 2015*). One million paired-end reads were used for each sample, providing enough depth to reach species-level resolution (*Hillmann et al., 2018*).

## Statistical analysis

Data presented were analyzed and visualized using the R statistical software (*R Core Team, 2017*). Statistical tests were used as described in the main text. For nonparametric statistical tests, multiple comparisons were performed using Dunn's test following Kruskal-Wallis using the *FSA* R package (*Ogle, 2018*), and corrected for multiple comparisons using Bonferonni correction. For many-to-one comparisons (*e.g.,* pharmacologic treatments compared to untreated controls), multiple hypothesis testing correction was accomplished by using Dunnett's test, implemented with the *multcomp* R package (*Hothorn et al., 2008*). For multiple comparisons between experimental groups, Tukey's honest significant difference (HSD) was used to correct for multiple testing. Unless otherwise noted, figures depict individual samples as points, and the bars indicate the median or mean ± SEM. In figures, *p < 0.05, ** p < 0.01, and ***p < 0.001.

## Repeated sampling of gnotobiotic mice

For the experiments in which gnotobiotic mice were used to assess the roles of host carrying capacity and microbiota fitness in shaping microbiota density, mice were sampled longitudinally to increase sample size for each condition. For the mice colonized with fecal samples from the lion, elephant, ferret, and red panda, two-way ANOVA shows that the main effect is the microbiota used to colonize the mouse (F = 32.3, p = $8.27 \times 10^{-16}$), while the identity of the individual mice does not contribute to the effects (F = 1.08, p = 0.388). The same is true for the mice colonized with fecal samples from individuals with IBD and pouch (F = 29.4, p < 0.0001 for the colonizing microbiota; F = 0.746, p = 0.634; two-way ANOVA). As a result, we are able to effectively measure the

microbiota density of gnotobiotic mice in these conditions and increase the utility of each gnotobiotic mouse.

## Supplemental results

### DNase inactivation buffer vs phenol chloroform DNA extraction comparison

To test if the two DNA extraction methods affected the resulting microbiota composition data, we processed separate aliquots from the same fecal sample using both methods. We found that the abundances of taxa in the sample processed with both methods were highly correlated (*Figure 1—figure supplement 1A and B*), suggesting that they represent equivalent ways to assay microbial community composition. In practice, the DIB method was most conducive to the small feces produced by mice and the large majority of mouse samples for this study were processed using this protocol, since the protocol utilizes smaller tubes that can be arrayed into standard 96-well formats. For the remaining mammals, the phenol:chloroform method was used as the number of stools used in the study was less, and the larger stools were more practical to aliquot into the wider 2.0 mL tube used for the phenol:chloroform method.

One possible limitation of using DNA content as a measurement of microbiota density is that small amounts of fecal matter contain sufficient DNA to saturate or clog the DNA binding columns used during extraction. This upper limit can largely be avoided by limiting the amount of input fecal material of higher microbiota density mammals (*e.g.,* mice) to < 50 mg and lower microbiota density mammals (*e.g.,* humans) to < 200 mg. In our experience, bead beating also becomes inefficient at >200 mg of fecal material. In contrast to the phenol:chloroform method, the DIB extraction protocol relies on a subsampling step that provides an additional safeguard to ensure the DNA extraction does not saturate the capacity of the Qiagen DNA-binding columns. By sampling a fraction of the lysate, we can extend the upper limit of our extraction protocol. At the extreme, using a 5 μL subsample of the lysate can increase the dynamic range by a factor of 140, which in turn implies that we can measure microbiota density for samples containing up to 1.4 mg of DNA (140 * 10 μg binding capacity of columns). On the lower end of our dynamic range, dye-based methods (Qubit Hi-sensitivity) provide an accurate detection down to 0.2 ng.

### qPCR quantification of DNA origin

While the dynamic range of the DIB extraction method described above is typically sufficient for stool samples, which contain high densities of microbial DNA compared with other environments, we further extended the method with qPCR-based quantification of the V4 region of the 16S rRNA gene. Additionally, by utilizing DNA yield per fecal sample as a measure of microbiota density, we assume host DNA is a minor contributor to the total fecal DNA yield.

To quantify the amount of bacterial and mouse DNA in our samples, we targeted the V4 region of the bacterial 16S rRNA gene (*Relman et al., 1992*) and the mouse TNFα gene (*Nitsche et al., 2001*). qPCR reactions were performed in 20 μL reaction volumes with final primer concentrations of 200 nM, using KAPA SYBR FAST Master Mix (2x) ROX Low (Kapa Biosystems). The thermal cycling and imaging were performed on the ViiA 7 Real-Time PCR System (Thermo Fisher).

We quantified the amount of host vs bacterial DNA in several samples by qPCR, and evaluated the qPCR performance against spike-in controls with known combinations of mouse and bacterial DNA. We found that even amongst samples with low microbial density (*e.g.,* samples from mice treated with vancomycin), the DNA content is largely microbial (*Figure 1—figure supplement 1C*). We were also able to measure the presence of microbial DNA down to concentrations near 1 pg/μL (*Figure 1—figure supplement 1C*). This allows us to measure microbial density for samples with DNA as low as 100 pg (minimum concentration 1 pg/μL in a 100 μL elution volume). Coupled with the ability to subsample the lysate from our DNA extraction protocol, this allows us to measure microbiota density across 5 orders of magnitude for the phenol:chloroform method and 7 orders of magnitude with the DIB protocol.

### Antibiotic spectrum and *in vivo* activity

By combining measures of microbiota density with sequencing-based measures of gut microbiota composition, we can study the ability of antibiotics to act within the context of a complex microbial community. We examined the 16S rRNA gene sequencing data from our antibiotics experiments

(*Figure 2*) to study the effects of polymyxin B, which acts by binding to the bacterial outer membrane that is present in gram-negative but not gram-positive organisms, and of vancomycin, which acts by inhibiting cell wall synthesis in gram-positive bacteria, and is thought to have little or no efficacy against gram-negative organisms. We focused on the changes in absolute abundances of bacterial phyla that are largely gram-positive (Actinobacteria and Firmicutes) or largely gram-negative (Bacteroidetes and Proteobacteria). Polymyxin B did not reduce the microbiota density overall, and did not significantly reduce the absolute abundance of Gram-negative bacteria ($p = 0.116$, Wilcoxon rank sum test; *Figure 2—figure supplement 1A*) or change the absolute abundance of Gram-positive bacteria ($p = 0.273$, Wilcoxon rank sum test, *Figure 2—figure supplement 1B*). Vancomycin, on the other hand, drove a significant decrease in the absolute abundance of both gram-positive and gram-negative organisms ($p_{Gram(+)} = 6.27 \times 10^{-13}$, $p_{Gram(-)} = 6.27 \times 10^{-13}$, Wilcoxon rank sum test; *Figure 2—figure supplement 1C and D*). These results suggest that the spectrum of activity of antibiotics as determined by *in vitro* assays may not reflect the effects of these drugs *in vivo*, when they are introduced to complex communities of organisms such as in the gut.

## Absolute microbial dynamics and alpha diversity in response to pharmacologics

Culture-independent measurements have revealed that antibiotics can disrupt the composition of a healthy gut microbiota (*Dethlefsen et al., 2008*). We hypothesized that antibiotics may also have an impact on the gut microbiota density. To test this hypothesis, we administered vancomycin in two doses (0.2 mg/mL and 0.5 mg/mL) to two sets of SPF C57BL/6J mice and collected fecal pellets before and during treatment. We found that vancomycin exerted selective pressure against susceptible organisms leading to a relative expansion of Verrucomicrobia and Firmicutes in the low and high dose groups respectively (*Figure 2—figure supplement 2A and B* ). When we multiplied each taxa's relative abundance by the microbiota density to calculate their absolute abundances, we observed a bloom of Verrucomicrobia in the low dose group (*Figure 2—figure supplement 2C* ). Surprisingly, in the high dose group, we found that vancomycin successfully depleted members of all phyla, including Firmicutes (*Figure 2—figure supplement 2D* ). Microbiota density and alpha diversity were not significantly correlated ($\rho = 0.107$; $p = 0.557$; Spearman; *Figure 2—figure supplement 2E*), as both low dose and high dose vancomycin significantly reduced alpha diversity ($p_{low} = 6.10 \times 10^{-5}$ and $p_{high} = 0.00223$, final timepoint vs baseline, Mann Whitney; *Figure 2—figure supplement 2F* ), while only high dose vancomycin reduced microbiota density ($p_{low} = 0.669$, $p_{high} = 0.0127$, final timepoint vs baseline, Mann Whitney; *Figure 2—figure supplement 2G* ).

## Identifying fungi in rCDI samples

Recent work has demonstrated that the fungal community may play an important role in modulating response to FMT in patients with rCDI (*Zuo et al., 2018*). We sought to identify whether our cohort of individuals with rCDI had a significant fungal component to their microbiota. We performed shotgun metagenomic sequencing on fecal samples from patients prior to FMT (n = 15) and profiled the composition of the microbial community using MetaPhlAn2 (*Truong et al., 2015*). Using this approach, we were only able to identify fungal reads in one of the eighteen samples. In this sample, Saccharomyces cerevisiae was the only identified fungi and comprised 0.112% of the mapped reads, while the remaining 99.9% were mapped to bacteria, consistent with previous reports of fungal reads accounting for approximately 0.1% of the human gut metagenome (*MetaHIT Consortium et al., 2010*).

One possible limitation of this analysis is that the methods described here are not specifically designed to extract and measure fungal DNA, as they do not utilize lyticase or a heat lysis step as in other protocols (*Iliev et al., 2012*; *Huseyin et al., 2017*; *Sokol et al., 2017*; *Tang et al., 2015*). Nonetheless, previous work by *Yu and Morrison (2004)* demonstrated that a bead-beating plus phenol:chloroform extraction method (*Whitford et al., 1998*), similar to the one employed for the rCDI samples used in this study was able to extract more DNA from rumen samples than other methods such as the QIAamp DNA Stool Mini Kit that have been employed in other fungal microbiome studies (*Suhr et al., 2016*). Furthermore, the identification of fungal reads in our samples demonstrates our methods were capable of isolating at least a proportion of the fungal DNA.

Another possible limitation of this approach is that the databases of published fungal genomes are relatively sparse. This limitation is shared among all sequencing-based approaches aimed at studying the fungal microbiome, and makes it possible that the real fungal fraction of the microbiome is larger than what we are able to identify.

## Acknowledgements

We are grateful to C Fermin, E Vazquez, and G Escano in the Mount Sinai Immunology Institute Gnotobiotic facility for their help with gnotobiotic animal husbandry. D Present and S Petrunio provided helpful suggestions during the course of this work. Next generation sequencing was performed at NYU School of Medicine by the Genome Technology Center partially supported by the Cancer Center Support Grant, P30CA016087. Human microbiome processing was performed in part by the Human Immune Monitoring Center at the Icahn School of Medicine at Mount Sinai. This work was supported in part by the staff and resources of Scientific Computing and of the Flow Cytometry Core at the Icahn School of Medicine at Mount Sinai. Raw sequencing files (fastq) for all 16S rRNA gene amplicon sequencing samples and metagenomic sequencing samples are stored in the public Sequence Read Archive (SRA) under project number PRJNA413199. Flow cytometry data is available through Mendeley Data (http://dx.doi.org/10.17632/cjvfrbyxhj.1).

## Additional information

### Competing interests

Ruiqi Huang, Marla Dubinsky, Jeremiah J Faith: Is a consultant for Janssen and has no other financial competing interests to declare. The other authors declare that no competing interests exist.

### Funding

| Funder | Grant reference number | Author |
|---|---|---|
| National Institute of Diabetes and Digestive and Kidney Diseases | DK112679 | Eduardo J Contijoch |
| National Institute of General Medical Sciences | GM108505 | Jeremiah J Faith |
| Leona M. and Harry B. Helmsley Charitable Trust | | Iris Dotan |
| Janssen Research and Development | | Eric E Schadt Marla Dubinsky Jeremiah J Faith |
| National Institute of General Medical Sciences | GM007280 | Eduardo J Contijoch Sean R Llewellyn |

The sampling of the Inflammatory Bowel Disease cohort (Crohn's disease and ulcerative colitis) was jointly designed by a collaborative effort between Mount Sinai and Janssen Research and Development. Beyond this exception, no other funders had no role in study design, data collection and interpretation, or the decision to submit the work for publication.

### Author contributions

Eduardo J Contijoch, Conceptualization, Data curation, Software, Formal analysis, Funding acquisition, Validation, Investigation, Visualization, Methodology, Writing—original draft, Writing—review and editing; Graham J Britton, Chao Yang, Investigation, Methodology, Writing—review and editing; Ilaria Mogno, Methodology, Writing—review and editing; Zhihua Li, Methodology; Ruby Ng, Sheela Hira, Revital Barkan, Robert P Hirten, Shih-Chen Fu, Yuying Luo, Nancy Yang, Tramy Luong, Philippe R Labrias, Roman Kosoy, Seunghee Kim-Schulze, Xiaochen Qin, Anabella Castillo, Amanda Hurley, Ashish Atreja, Jason Rogers, Farah Fasihuddin, Merjona Saliaj, Amy Nolan, Pamela Reyes-Mercedes, Carina Rodriguez, Sarah Aly, Kenneth Santa-Cruz, Mayte Suárez-Fariñas, Ruiqi Huang, Ke Hao, Jun

Zhu, Bin Zhang, Bojan Losic, Won-Min Song, Antonio Di Narzo, Wenhui Wang, Benjamin L Cohen, Christopher DiMaio, David Greenwald, Steven Itzkowitz, Aimee Lucas, James Marion, Elana Maser, Ryan Ungaro, Steven Naymagon, Joshua Novak, Brijen Shah, Thomas Ullman, Peter Rubin, James George, Peter Legnani, Joshua R Friedman, Carrie Brodmerkel, Scott Plevy, Resources; Sean R Llewellyn, Investigation, Methodology; Crystal Johnson, Andrew Kasarskis, Resources, Supervision, Writing—review and editing; Keren M Rabinowitz, Lauren Peters, Haritz Irizar, Shannon E Telesco, Resources, Writing—review and editing; Iris Dotan, Eric E Schadt, Resources, Supervision, Funding acquisition; Sergio Lira, Inga Peter, Judy H Cho, Resources, Funding acquisition; Ari Grinspan, Carmen Argmann, Marla Dubinsky, Bruce Sands, Resources, Supervision; Jose C Clemente, Jean-Frederic Colombel, Resources, Supervision, Funding acquisition, Writing—review and editing; Jeremiah J Faith, Conceptualization, Resources, Supervision, Funding acquisition, Writing—original draft, Project administration, Writing—review and editing

### Author ORCIDs

Eduardo J Contijoch http://orcid.org/0000-0002-2873-6808
Jose C Clemente http://orcid.org/0000-0002-3970-9856
Antonio Di Narzo http://orcid.org/0000-0002-4033-5038
Judy H Cho http://orcid.org/0000-0002-7959-0466
Jeremiah J Faith http://orcid.org/0000-0002-2691-4500

### Ethics

Human subjects: Informed consent was obtained from all human subjects, and the Mount Sinai Institutional Review Board approved this study. Individual ages 18 and over were recruited to be part of the study using a protocol approved by the Mount Sinai Institutional Review Board (HS# 11-01669). Once the coordinators went over the consent form and subjects consented to be part of the study to be published with subjects deidentified, they were given a study identification number that all their study samples were labeled with. All study samples were processed with no identifiers linked to them other than their study id.

Animal experimentation: All animal experiments in this study were approved by Institutional Animal Care and Use Committee (IACUC) of the Icahn School of Medicine (protocol: IACUC-2013-1385) and were performed in accordance with the approved guidelines for animal experimentation at the Icahn School of Medicine at Mount Sinai.

### Decision letter and Author response

Decision letter https://doi.org/10.7554/eLife.40553.032
Author response https://doi.org/10.7554/eLife.40553.033

# Additional files

### Supplementary files

• Supplementary file 1. Mammalian sample information. This table contains information on the mammalian species used in this study, including taxonomic information, diet, and approximate mass.
DOI: https://doi.org/10.7554/eLife.40553.024

• Supplementary file 2. Antibiotics used in mouse experiments. This table contains information on the antibiotics in this study, including the concentrations used to treat mice and the sources used to determine the final dosing.
DOI: https://doi.org/10.7554/eLife.40553.025

• Transparent reporting form
DOI: https://doi.org/10.7554/eLife.40553.026

### Data availability

Raw sequencing files (fastq) for all 16S sequencing samples and shotgun metagenomic sequencing are stored in the public Sequence Read Archive (SRA) under project number PRJNA413199.

The following datasets were generated:

| Author(s) | Year | Dataset title | Dataset URL | Database and Identifier |
|---|---|---|---|---|
| Eduardo J Contijoch, Graham J Britton, Chao Yang, Ilaria Mogno, Zhihua Li, Ruby Ng, Sean R Llewellyn, Sheela Hira, Crystal Johnson, Keren M Rabinowitz, Revital Barkan, Iris Dotan, Robert P Hirten, Shih-Chen Fu, Yuying Luo, Nancy Yang, Tramy Luong, Philippe R Labrias, Sergio Lira, Inga Peter, Ari Grinspan, Jose C Clemente, Roman Kosoy, Seunghee Kim-Schulze, Xiaochen Qin, Anabella Castillo, Amanda Hurley, Ashish Atreja, Jason Rogers, Farah Fasihuddin, Merjona Saliaj, Amy Nolan, Pamela Reyes-Mercedes, Carina Rodriguez, Sarah Aly, Kenneth Santa-Cruz, Lauren Peters, Mayte Suárez-Fariñas, Ruiqi Huang, Ke Hao, Jun Zhu, Bin Zhang, Bojan Losic, Haritz Irizar, Won-Min Song, Antonio Di Narzo, Wenhui Wang, Benjamin L Cohen, Christopher DiMaio, David Greenwald, Steven Itzkowitz, Aimee Lucas, James Marion, Elana Maser, Ryan Ungaro, Steven Naymagon, Joshua Novak, Brijen Shah, Thomas Ullman, Peter Rubin, James George, Peter Legnani, Shannon E Telesco, Joshua R Friedman, Carrie Brodmerkel, Scott Plevy, Judy H Cho, Jean-Frederic Colombel, Eric E Schadt, Carmen Argmann | 2018 | Data Related to: Gut microbiota density influences host physiology and is shaped by host and microbial factors | http://dx.doi.org/10.17632/cjvfrbyxhj.1 | Mendeley Data, 10.17632/cjvfrbyxhj.1 |
| Eduardo J Contijoch, Graham J Britton | 2018 | The absolute gut microbiome alters host physiology, varies by gut architecture and disease, and predicts response to therapy | https://www.ncbi.nlm.nih.gov/bioproject/PRJNA413199 | NCBI BioProject, PRJNA413199 |

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
