## [Decision Letter]

[**Editorial note:** This article has been through an editorial process in which the authors decide how to respond to the issues raised during peer review. The Reviewing Editor's assessment is that all the issues have been addressed.]

Thank you for submitting your article "Gut microbiota density influences host physiology and is shaped by host and microbial factors" for consideration by *eLife*. Your article has been reviewed by two peer reviewers, one of whom is a guest Reviewing Editor, and the evaluation has been overseen by Wendy Garrett as the Senior Editor. The reviewers have opted to remain anonymous.

The Reviewing Editor has highlighted the concerns that require revision and/or responses, and we have included the separate reviews below for your consideration. If you have any questions, please do not hesitate to contact us.

Summary:

In this study, the authors evaluated the microbiota density in fecal samples using DNA quantification in different clinical settings and mouse experiments. The results suggest that microbiota density is an important parameter to take into account.

Major concerns:

Major concerns have been raised by reviewer 1 regarding the methodology used to assess microbiota density, analysis of Clostridium difficile infection samples and several other aspects. These issues should be addressed by authors before potential publication.

Separate reviews (please respond to each point):

*Reviewer #1:*

In this study Contijoch and colleagues aimed at evaluating the microbiota density in fecal sample and its potential consequences in IBD and fecal microbiota transplantation in rCDI. For these purpose, the authors used samples from several animal and human cohorts as well as performed FMT experiments in germ free mice.

The topic is highly relevant. However there are major limitations impairing the scientific contribution of this study. I do not think that the parameter defined by the author is appropriate to define the microbiota density.

Major comments:

– The authors studied only the bacterial fraction of the microbiota. Although present in smaller number, other microorganisms such a fungi car represent a significant portion of the biomass, specifically in antibiotics treated animal. Is the DNA extraction method used appropriate to quantify fungi? This is particularly important as a fungi as a volume which is 100 fold bigger than a bacteria. This also mean tha using the DNA quantification is not appropriate if one want to quantify the microbiota biomass. This should be at least discussed.

– The authors looked at the effects of different antibiotics. It would be interesting to correlate the obtained results with the spectrum of action of these antibiotics and with microbiota composition.

– Assuming that the variation in microbiota density is not linked to variation in water content, what is the component (increased or decreased) explaining this variation in density?

– Results section: "These results from measuring the absolute microbiota differ from common observations of relative increases in Proteobacteria associated with IBD". I am not sure the author can write "differ" as they are looking at absolute concentration while cited worked looked at proportion. So these results are not contradictory.

– For the Cdiff study: when was the sample of the patient collected regarding antibiotics (vancomycin or other) intake ? If this patients were on (or recently on) antibiotics, these results are useless as this represent a major cofounding factor (specially based on the results showed in Figure 2).

– For the Cdiff Germ-free mice experiment: I agree that the decreased microbiota density in germfree mice colonized with rCDI microbiota is compatible with a lower fitness. However the correction of the microbiota density following FMT might be just related to the input of other bacterial species more adapted to the colonic environment. So this results is very likely to be related to the type of bacteria present. So I do not really see the point of the authors here. In any case, the data exposed do not support the role of microbiota density independently of the microbiota quality.

Minor comments:

– It is not clear why the authors used Pearson correlation in some Figure (Figure 1B for example) and Spearman correlation in other. Given the type of data analyzed, it is probably safer to only use Spearman correlation.

– How do the authors explain that Ciprofoxacin or Metronidazole alone decrease microbiota density but the association Ciprofoxacin + Metronidazole does not?

*Reviewer #2:*

I do not have substantive concerns or requests for additional work for this paper.

The authors provide evidence that microbiota density varies between gut ecologies and disease states, and can be corrected by FMT. The paper includes cross-species comparisons, studies of humans in healthy and disease states, and experimental approaches. Together, these studies introduce a microbiota density measurement approach that is readily integrated into standard analysis pipelines and establish important relationships between gut anatomy, fecal water content, and microbiota density. The authors further provide a panel of treatments (laxatives, antibiotics) that increase or decrease microbiota density that will be of use to many in the field. Additionally, the authors provide evidence that "blooms" of Proteobacteria often associated with dysbiosis may in cases instead reflect Proteobacterial stability during decreases of other taxa. This is an important concept that could change the way the field thinks about dysbiosis.

Minor Comments:

If space permits, the authors could include a section in the Discussion covering potential mechanisms/reasons for the relationship between microbiota density and disease.

The Results section, second paragraph, states that lion and red panda microbiota reach higher densities in the mouse than in the native host, suggesting their densities were limited by the carrying capacity of their native host. It could be helpful to clarify this conclusion – is it possible that the limitation is due to the native host's diet or environment (either in the zoo or in general), rather than the animal's physiology?

---

## [Author Response]

Reviewer #1:

In this study Contijoch and colleagues aimed at evaluating the microbiota density in fecal sample and its potential consequences in IBD and fecal microbiota transplantation in rCDI. For these purpose, the authors used samples from several animal and human cohorts as well as performed FMT experiments in germ free mice.The topic is highly relevant. However there are major limitations impairing the scientific contribution of this study. I do not think that the parameter defined by the author is appropriate to define the microbiota density.Major comments:– The authors studied only the bacterial fraction of the microbiota. Although present in smaller number, other microorganisms such a fungi car represent a significant portion of the biomass, specifically in antibiotics treated animal. Is the DNA extraction method used appropriate to quantify fungi? This is particularly important as a fungi as a volume which is 100 fold bigger than a bacteria. This also mean tha using the DNA quantification is not appropriate if one want to quantify the microbiota biomass. This should be at least discussed.

The reviewer makes two important points here first how to broaden the applicability of the methods described in this manuscript to include microbes beyond bacteria (which our initial analyses were limited to because of the focus on 16S rRNA sequencing to assay community composition) and second what specifically are we measuring by this metric, as differences in the scale of these organisms would require adjustments for metrics that aim to reflect the volume occupied by the organisms in communities that contain a mixture of viruses, fungi, and bacteria where all three are being studied in parallel.

Although the investigation of the fungal component of the microbiome was not a primary focus of our study, we have updated the manuscript to include a discussion of the fungal microbiome and strategies to address this issue in our current work and moving forward. To specifically address the contributions of fungi in a subset of human samples most likely to be affected by antibiotic usage, we performed metagenomic sequencing on a subset of the individuals with recurrent Clostridium difficile infection (rCDI) prior to fecal microbiota transplantation. Although our DNA extraction protocol is not the optimal published method for extracting fungi, in a published comparison it performed similar to more fungi specific protocols (Yu and Morrison, 2004). However in all tested human rCDI samples, fungi represented only a minor proportion of each sample <0.1%.

As demonstrated in the manuscript (e.g., Figure 1—figure supplement 1D), the DNA based measurement of microbiota density employed in this manuscript best corresponds to a culture independent version of a colony forming unit (CFU). In this context, fungi and bacteria should both be well characterized by this metric as colony-forming units are similar to the “genome equivalents” expressed by metagenomic algorithms that correct for the differences in genome size between organisms. However, we agree that if the goal of other absolute metrics of gut microbiota composition is to specifically quantify the volume represented by each organism rather than the countable number of each organism in a sample, then corrections would need to be made using the established volumes of each strain of interest.

We believe all of these are important points to more specifically understand the microbiota density metric used in this manuscript and have thus include a more complete discussion of these topics in the Discussion:

“Although bacteria dominate the gut microbiota and were the primary focus of this study, this approach could also be used to account for the non-bacterial (e.g., fungal, viral, protozoan) components of the microbiome, assuming an appropriate DNA extraction method was chosen to efficiently lyse these broader microbial groups. While recent studies that examined the fungal microbiome have shown that fungi can alter the efficacy of FMT in rCDI for selected individuals (Zuo et al., 2018), we did not detect a substantial contribution of fungi in the metagenome of our rCDI samples extracted with phenol:chloroform and bead beating (see Supplemental Results). As described above, microbial density is highly correlated with CFU and ultimately each microbial genome is associated with a single organism. Therefore, as long as the relative abundance measure is scaled by the size of each organism’s genome (i.e., genome equivalents) the density of each species estimated by this DNA-based method should also roughly reflect the CFU density of each species with the same caveats of 16S rRNA copy number and genome copy number that apply to all microbiome studies. As measures of microbiota density seek to become more accurate, well-curated databases that contain information on fundamental characteristics of microbes such as cell volume or mass could provide further refinement.”

And in our section titled “Identifying fungi in rCDI samples”:

“Recent work has demonstrated that the fungal community may play an important role in modulating response to FMT in patients with rCDI (Zuo et al., 2018). We sought to identify whether our cohort of individuals with rCDI had a significant fungal component to their microbiota. We performed shotgun metagenomic sequencing on fecal samples from patients prior to FMT (n = 15) and profiled the composition of the microbial community using MetaPhlAn2 (Truong et al., 2015). Using this approach, we were only able to identify fungal reads in one of the eighteen samples. In this sample, *Saccharomycescerevisiae* was the only identified fungi and comprised 0.112% of the mapped reads, while the remaining 99.9% were mapped to bacteria, consistent with previous reports of fungal reads accounting for approximately 0.1% of the human gut metagenome (Qin et al., 2010).

One possible limitation of this analysis is that the methods described here are not specifically designed to extract and measure fungal DNA, as they do not utilize lyticase or a heat lysis step as in other protocols (Iliev et al., 2012; Sokol et al., 2017; Tang et al., 2015). Nonetheless, previous work by Yu and Morrison (2004) demonstrated that a bead-beating plus phenol:chloroform extraction method (Whitford et al., 1998), similar to the one employed for the rCDI samples used in this study was able to extract more DNA from rumen samples than other methods such as the QIAamp DNA Stool Mini Kit that have been employed in other fungal microbiome studies (Suhr et al., 2016). Furthermore, the identification of fungal reads in our samples demonstrates our methods were capable of isolating at least a proportion of the fungal DNA.

Another possible limitation of this approach is that the databases of published fungal genomes are relatively sparse. This limitation is shared among all sequencing-based approaches aimed at studying the fungal microbiome, and makes it possible that the real fungal fraction of the microbiome is larger than what we are able to identify.”

– The authors looked at the effects of different antibiotics. It would be interesting to correlate the obtained results with the spectrum of action of these antibiotics and with microbiota composition.

We have investigated the relationship between spectrum of antibiotic activity and changes in absolute abundance of microbes. However, these results are limited by the difficulty in reconciling clinical classifications of bacteria based on biochemical characteristics such as gram staining and the phylogenetic classification of bacteria that rely upon sequence similarities. We present data for vancomycin, which is known to cover gram-positive organisms and not gram-negative organisms, and polymyxin B, which is known to cover gram-negative organisms and not gram-positive organisms, and their effects on the abundances of bacterial phyla that are largely gram-negative (Bacteroidetes and Proteobacteria) and gram-positive (Actinobacteria and Firmicutes). Despite the somewhat focused spectrum of these antibiotics *in vitro*, we find *in vivo* that each antibiotic’s microbiota depletion capability did not align well with antibiotic spectrum. Vancomycin efficiently removed both gram positive and negative organisms, while polymyxin B had little effect on the microbiota. The potential reasons for broad efficacy of vancomycin could be that it slows the grow of gram-negative bacteria to the point that they cannot replicate faster than the transit time of the host or that collapsing the gram-positive component of the microbiota eliminates key nutrients and synergistic partners necessary for sustaining high levels of gram negative organisms.

These new results are included in the section titled “Antibiotic spectrum and *in vivo* activity”:

“By combining measures of microbiota density with sequencing-based measures of gut microbiota composition, we can study the ability of antibiotics to act within the context of a complex microbial community. We examined the 16S rRNA gene sequencing data from our antibiotics experiments (Figure 2) to study the effects of polymyxin B, which acts by binding to the bacterial outer membrane that is present in gram-negative but not gram-positive organisms, and of vancomycin, which acts by inhibiting cell wall synthesis in gram-positive bacteria, and is thought to have little or no efficacy against gram-negative organisms. We focused on the changes in absolute abundances of bacterial phyla that are largely gram-positive (Actinobacteria and Firmicutes) or largely gram-negative (Bacteroidetes and Proteobacteria). Polymyxin B did not reduce the microbiota density overall, and did not significantly reduce the absolute abundance of Gram-negative bacteria (p = 0.116, Wilcoxon rank sum test; Figure 2—figure supplement 1A) or change the absolute abundance of Gram-positive bacteria (p = 0.273, Wilcoxon rank sum test, Figure 2—figure supplement 1B). Vancomycin, on the other hand, drove a significant decrease in the absolute abundance of both gram-positive and gram-negative organisms (p_Gram(+)_ = 6.27 x 10^-13^, p_Gram(-)_ = 6.27 x 10^-13^, Wilcoxon rank sum test; Figure 2—figure supplement 1C and 1D). These results suggest that the spectrum of activity of antibiotics as determined by *in vitro* assays may not reflect the effects of these drugs *in vivo*, when they are introduced to complex communities of organisms such as in the gut.”

– Assuming that the variation in microbiota density is not linked to variation in water content, what is the component (increased or decreased) explaining this variation in density?

The reduction in the density is tied to a reduction in the number of microbes in the gut, regardless of if those organisms are in a more water rich or dry matter rich environment. We attempted in numerous ways throughout the manuscript to address the possibility that certain samples could have a reduced microbiota density due solely to an increase in fecal water content, and we continually found that water content alone does not explain the variation in microbiota density. The component that explains the variation in density is the microbial content itself, as reflected by the correlation we find with microbiota density and culture-dependent quantification methods (Figure 1—figure supplement 1D). When we specifically focused on identifying contributions to fecal water content, we found that diet was one of the most important factors. We have previously shown that increasing dietary protein significantly increases microbiota density (Llewellyn et al., 2018), however we now demonstrate that this large increase in microbiota density results in stools with higher water content (the opposite of what would be expected if water dilution was the main factor driving microbiota density; see Figure 2—figure supplement 4). We found dietary fiber type was a strong predictor of fecal water content as we observe significant ~4-fold increases in water content when switching from a non-soluble to a soluble fiber (cellulose and psyllium respectively).

We have addressed this in the manuscript in our Discussion section:

“Our results suggest that microbiota density may change independently of water content, implying that the density of microbes can be altered independently of water and other contents of the stool bulk, such as undigested dietary components or host tissue.”

– Results section: "These results from measuring the absolute microbiota differ from common observations of relative increases in Proteobacteria associated with IBD". I am not sure the author can write "differ" as they are looking at absolute concentration while cited worked looked at proportion. So these results are not contradictory.

We agree that our language was not precise enough and might suggest that our results are in conflict with the previous results. As the reviewer points out, our results are not contradictory, but complementary to prior work that looked at relative abundances. Our data show that the absolute abundance of Proteobacteria is not different in individuals with IBD compared to healthy individuals, even though we observe an increase the relative abundance of Proteobacteria.

The text has been clarified to more precisely reflect the distinction:

“These results from measuring the density of each phyla provide a novel insight compared to previous studies that associated a relative increase in the proportion of Proteobacteria with IBD (Frank et al., 2007; Gevers et al., 2014). We show here that in absolute terms, Proteobacteria are able to sustain a constant density in the individuals with IBD while the remaining phyla decrease in density.”

– For the Cdiff study: when was the sample of the patient collected regarding antibiotics (vancomycin or other) intake ? If this patients were on (or recently on) antibiotics, these results are useless as this represent a major cofounding factor (specially based on the results showed in Figure 2).

As pointed out by the reviewer, we show in this paper that antibiotic treatment can be a significant contributing factor to microbiota density (e.g., Figure 2), which could confound analysis of rCDI individuals taking antibiotics. We therefore sought to control for the use of antibiotics by employing a gnotobiotic mouse model of FMT where mice colonized with the microbiota of individuals with rCDI are not given antibiotics and are treated with FMT. These results show that the microbiota of individuals with rCDI has reduced microbiota fitness compared to healthy individuals because when controlling for the host and antibiotic exposure factors, mice colonized with the rCDI microbiota have reduced microbiota density compared to mice colonized with the microbiota of healthy individuals.

We updated the text of the manuscript to highlight the point raised by the reviewer:

“However, these results may be confounded by the fact that the individuals with rCDI have been exposed to antibiotic treatment prior to their FMT, and as we showed in Figure 2A, antibiotics may reduce microbiota density.

To separate the host physiologic and pharmacologic factors that might impact our understanding of community fitness in rCDI, we utilized a gnotobiotic murine model of FMT (Figure 4D) where germ-free mice were initially colonized with the fecal material of individuals with rCDI for 3 weeks prior to a single transplant of fecal material via oral gavage from a second human donor – the same healthy FMT donor used for the transplant clinically. The ex-germ-free mice therefore model the fecal microbiota transplant but in a fixed environment, with a controlled diet, and no antibiotic confounder.”

– For the Cdiff Germ-free mice experiment: I agree that the decreased microbiota density in germfree mice colonized with rCDI microbiota is compatible with a lower fitness. However the correction of the microbiota density following FMT might be just related to the input of other bacterial species more adapted to the colonic environment. So this results is very likely to be related to the type of bacteria present. So I do not really see the point of the authors here. In any case, the data exposed do not support the role of microbiota density independently of the microbiota quality.

It appears we are in agreement on all points here. We also think the species composition of the rCDI is suboptimal for a gut and thus does not grow to high density even in an antibiotic free polysaccharide rich diet environment that should facilitate high densities. This implies that the community contains members that are individually or collectively suboptimal in their fitness for the colon. In the manuscript, we focus on two forces driving microbiota density either those from the host (i.e., host carrying capacity) or those from the microbes themselves (i.e., microbiota fitness). In the context of rCDI, it appears that the microbes themselves are unfit (i.e., their quality contributes to their density) and when a more fit community is transplanted into the recipient it reestablishes a community at a higher density than the prior one. The microbiota density measurement provides a quantitative way to determine this loss of fitness and its restoration. It provides some mechanistic insights into how FMT might work for rCDI (and perhaps not work as easily in other conditions). Although these results are intuitive in rCDI, quantifying them is important as it allows the study of responders and non-responders to FMT, it might enable a more precise understanding of FMT in other indications, and it will facilitate identifying at-risk individuals with unfit microbiota that might make them less able to have colonization resistance to other pathogens.

We updated the text of the manuscript to include this in our Discussion:

“Moving forward, studying the factors that determine both host carrying capacity and microbiota fitness may allow us to predict which disease states may benefit from therapeutics that target the host versus ones that target the microbiota. By identifying components of the microbiota that confer increased fitness, we can improve our understanding of the ecological rules that govern the microbiome. For example, exploring how FMT is able to increase microbiota fitness and therefore microbiota density should provide mechanistic insights into FMT for rCDI that can be used for other potential indications for FMT. These results also suggest that routine monitoring could identify individuals with microbiota fitness deficiencies that might benefit from prophylactic microbial therapeutics to boost colonization resistance to treat or prevent disease (Battaglioli et al., 2018).”

Minor comments:– It is not clear why the authors used Pearson correlation in some Figure (Figure 1B for example) and Spearman correlation in other. Given the type of data analyzed, it is probably safer to only use Spearman correlation.

The challenge with using Spearman in this context is the dataset has a sampling bias towards mammals that are phylogenetically diverse, which we purposefully did to obtain a broad look at microbiota density across mammals. Using a parametric statistic better captured that the phylogenetically similar organisms (e.g., primates, carnivores, rodents, etc.) also had similar microbiota densities. However, to be consistent with the rest of the manuscripts use of non-parametric statistics, we have removed this result, which is not significant with Spearman. However, a similar observation can be made and overcome the obstacle of unbalanced sampling of phylogentically similar and dissimilar organisms by grouping the organisms at the level of order. Similar to the phylogenetic distance based metric suggesting organisms that are more genetically related have more similar microbiota densities, we find that organisms in the same order have more similar microbiota densities by non-parametric Kruskal-Wallis test.

We have changed the text of manuscript to address the reviewer’s points:

“At the higher taxonomic rank of order, where we sampled at least two unique species (*Atriodactyla, Carnivora, Primates*, and *Rodentia*), we still found significant differences in microbiota density (H = 39.0, p = 3.39 x 10^-9^; Kruskal-Wallis), suggesting that evolutionarily conserved host features impact microbiota density. We found no correlation between microbiota density and either fecal water content (ρ= -0.0418, p = 0.892, Spearman; Figure 1B) or host size (mass) (ρ= -0.364, p = 0.167, Spearman; Figure 1—figure supplement 2)”

– How do the authors explain that Ciprofoxacin or Metronidazole alone decrease microbiota density but the association Ciprofoxacin + Metronidazole does not?

We do not have a good explanation for this other than that sometimes the combined effect of drugs is not the sum of their individual effects. We feel this is a pretty minor result of the manuscript that does not change any of the key conclusions of the manuscript.

Reviewer #2:

I do not have substantive concerns or requests for additional work for this paper.The authors provide evidence that microbiota density varies between gut ecologies and disease states, and can be corrected by FMT. The paper includes cross-species comparisons, studies of humans in healthy and disease states, and experimental approaches. Together, these studies introduce a microbiota density measurement approach that is readily integrated into standard analysis pipelines and establish important relationships between gut anatomy, fecal water content, and microbiota density. The authors further provide a panel of treatments (laxatives, antibiotics) that increase or decrease microbiota density that will be of use to many in the field. Additionally, the authors provide evidence that "blooms" of Proteobacteria often associated with dysbiosis may in cases instead reflect Proteobacterial stability during decreases of other taxa. This is an important concept that could change the way the field thinks about dysbiosis.Minor Comments:If space permits, the authors could include a section in the Discussion covering potential mechanisms/reasons for the relationship between microbiota density and disease.

Our results demonstrate a relationship between microbiota density and disease, but do not clearly indicate a causal relationship. However, our findings that manipulating microbiota density can alter host metabolism and immune cell populations provide a possible mechanistic link between microbiota density and disease. For example, in the case of IBD, it is unlikely that changes in microbiota density alone would cause disease, but it is possible that reduced microbiota density indicate a reduced microbiota fitness that may allow for the colonization of pathobionts that contribute to disease. Alternatively, the reduced microbiota density may contribute to a pro-inflammatory environment that can trigger a positive feedback loop to maintain microbiota density low in disease. The mechanisms that could link microbiota density to disease are complex, but represent a potentially fruitful area for future work.

We have updated our manuscript to address this comment:

“We also observed that microbiota density is reduced in individuals with IBD. Coupled with our findings that changes in microbiota density can alter host metabolism and immune populations, these results suggest that chronically low microbiota density may play a role in the development or progression of disease. It might even be possible that an initial reduction in microbiota density contributes to a pro-inflammatory host immune system that creates a positive feedback loop that sustains a low microbiota density. It is also possible that a low microbiota density, if due to low microbiota fitness, has reduced colonization resistance, allowing for pathogens or pathobionts to take hold and contribute to disease processes in the host (Battaglioli et al., 2018).”

The Results section, second paragraph, states that lion and red panda microbiota reach higher densities in the mouse than in the native host, suggesting their densities were limited by the carrying capacity of their native host. It could be helpful to clarify this conclusion – is it possible that the limitation is due to the native host's diet or environment (either in the zoo or in general), rather than the animal's physiology?

We suggest that the native host diet and environment should be considered features of the host that in their own way contribute to the host’s carrying capacity. Although the host carrying capacity may also be influenced by physiologic features that may not be readily altered or manipulated, there are certainly aspects of the host carrying capacity that can be modified (diet, social habits, etc.).

We have attempted to clarify this in the text:

“In germ-free Swiss Webster mice colonized with four of the lowest density microbiotas in our initial screen (lion, elephant, ferret, and red panda), the lion and red panda microbiotas reached higher microbiota densities in the mouse than in the native host, suggesting their densities were limited by the carrying capacity of their host (which could include factors like host diet and host social behaviors).”

And in the Discussion:

“The low microbiota density of the red panda, a member of *Carnivora* with a herbivorous diet, further supports intestinal architecture as a major determinant of host carrying capacity and thus a driver of microbiota density. Finally, the significantly reduced microbiota density in humans with IPAA uniquely demonstrates that changing gut architecture within a species (in this case by surgery to treat ulcerative colitis) is equally capable of influencing host carrying capacity. Outside of animal models, it is possible that other host features that may be more readily altered, such as dietary habits or social behaviors, may also influence the host carrying capacity.”